

# Speciation of anthropogenic emissions of non-methane volatile organic compounds: a global gridded data set for 1970-2012

Ganlin Huang[1], Rosie Brook[2], Monica Crippa[3], Greet Janssens-Maenhout[3], Christian Schieberle[1], Chris Dore[2], Diego Guizzardi[4], Marilena Muntean[3], Edwin Schaaf[3], Rainer Friedrich[1]

[1]Institute of Energy Economics and Rational Energy Use (IER), Universität Stuttgart, Hessbruehlstr. 49a, 70565, Stuttgart, Germany
[2]Aether, Emissions Inventory Consultancy, Oxford Centre for Innovation, New Road, Oxford OX1 1BY, United Kingdom
[3]European Commission, Joint Research Centre (JRC), Directorate for Energy, Transport and Climate, Air and Climate Unit, Via E. Fermi 2749, I-21027 Ispra (VA), Italy
[4]Didesk Informatica, Verbania (VB), Italy

*Correspondence to*: Ganlin Huang (Ganlin.huang@ier.uni-stuttgart.de)

**Abstract.** Non-methane volatile organic compounds (NMVOC) include a large number of chemical species which differ

significantly in their chemical characteristics and thus in their impacts on ozone and secondary organic aerosols formation. It is important that chemical transport models (CTMs) simulate the chemical transformation of the different NMVOC species in the troposphere consistently. In most emission inventories, however, only total NMVOC emissions are reported, which need to be decomposed into classes to fit the requirements of CTMs. For instance, the Emissions Database for Global Atmospheric Research (EDGAR) provides spatially resolved global anthropogenic emissions of total NMVOC. In this study

the EDGAR NMVOC inventory was revised and extended in time and in sectors. Moreover the new version of NMVOC emission data in the EDGAR database were disaggregated on a high sector resolution to individual species or species groups, thus enhancing the usability of the NMVOC emission data by the modelling community. Region- and source-specific speciation profiles of NMVOC species or species groups, are compiled and mapped to EDGAR processes (high resolution of sectors), with corresponding quality codes specifying the quality of the mapping. Individual NMVOC species in different

profiles are aggregated to 25 species groups, in line with the common classification of the Global Emissions Initiative (GEIA). Global annual grid maps with a resolution of $0.1\,°\times 0.1\,°$ for the period 1970-2012 are produced by sector and species. Furthermore, trends of NMVOC composition are analysed taking road transport and residential sources in Germany and the United Kingdom (UK) as examples.

## 1 Introduction

Non-methane volatile organic compounds (NMVOC) consist of a variety of chemical species which can give rise to increases in tropospheric ozone concentrations and the formation of secondary organic aerosols (EEA, 2015; Guenther et al.,



2012; Piccot et al., 1992). Some NMVOC species are toxic substances and can cause direct damage to human health (Weichenthal et al., 2012). A number of regulations, e.g. the Directive on ambient air quality and cleaner air for Europe (2008/50/EC), the Industrial Emissions Directive (2010/75/EU) and the Decopaint Directive (2004/42/EC), limit the emissions of NMVOC or the concentration of secondary pollutants, for example ozone (Theloke and Friedrich, 2007) and

particulate matter. Due to the non-linear relationship between emissions of NMVOC and concentrations of secondary pollutants formed in the atmosphere, chemical transport models (CTMs) are typically used to assess the effectiveness of potential air pollution control strategies and policies. Control strategies can take the form of abatement strategies or action plans, which lead to a reduction in ambient air concentrations, and achievement of target thresholds. Given the reactive nature of NMVOC in the atmosphere, it is important that CTMs simulate the chemical transformation of the different

NMVOC species in the troposphere to the best extent possible. To serve as input for chemistry models, the bulk NMVOC emissions need to be disaggregated to give information on species (or species groups) on a sector-by-sector basis. This is because different NMVOC species vary significantly in their chemical features and thus in their impacts on ozone and secondary organic aerosol formation. Whilst it is possible to consider the atmospheric chemistry of individual species, it is more practical for chemistry models to use species groups, which contain species similar in chemical structure or reactivity.

Determining and compiling NMVOC speciation profiles have attracted increasingly more scientific interests (Liu et al., 2008; Passant, 2002; Schultz et al., 2007; Theloke and Friedrich, 2007). However, all these studies have limited scope with regards to coverage in emission sources, species or target regions. In addition, these studies typically include speciation profiles that do not match the level of sectoral disaggregation at which total NMVOC emissions are typically reported in emission inventories. It is therefore challenging to collect NMVOC speciation profiles for different sources and regions, and

to map them to existing emission inventories. However, undertaking this task provides information on the species composition of total NMVOC emissions data, which would serve as input data for CTMs and related health impact assessments.

NMVOC emissions are typically reported in national emission inventories as total NMVOC, rather than individual species, or species groups. Although data exist on the emission of individual species, it is not a reporting requirement under

international conventions and therefore is difficult to collate. Generally, data at individual or grouped NMVOC level are provided by taking total NMVOC emissions of different emission sources from existing emission inventories and then applying speciation profiles. These speciation profiles represent the share of different NMVOC species, or species groups within the total NMVOC emissions. The occurrence and magnitude of individual species can vary considerably depending on the emission source, and it is therefore necessary to collate speciation profiles that are source- and fuel-specific. In

addition, NMVOC speciation profiles are expected to vary on a geographical basis, caused by differences in fuel quality, combustion technologies, and end-of-pipe control measures.

The Emissions Database for Global Atmospheric Research (EDGAR) provides spatially resolved global anthropogenic emissions of greenhouse gases and air pollutants. The total NMVOC emissions of EDGARv4.3.1 (Crippa et al., 2016) are reported, with no information on subdivisions into NMVOC species or species groups. This study updates the NMVOCs and



disaggregates (speciate) NMVOC emission data in the EDGAR database to individual species or species groups on the same sector resolution as the total NMVOC. Thus, the usability of the EDGAR data by the modelling community may be enhanced.

Region- and sector-specific speciation profiles are developed and provided. The most appropriate speciation profile for each EDGAR process (category of emission sources) is identified and mapped at subsector level for different regions. The species structure and different speciation profiles are unified by aggregating individual NMVOC species to species groups, according to their chemical structure and reactivity. Combining total NMVOC emissions and speciation profiles, speciated NMVOC emissions by country and global $0.1° \times 0.1°$ grid maps for different processes are generated, from which trends of NMVOC composition are assessed. Quality assessment of the generated data sets is performed. A comparison with other speciated NMVOC emission inventory data is also conducted and discussed.

## 2 Methodology and data

Our general approach is as follows: we start with a systematic literature review, searching speciation profiles from regional measurements and database, and apply the available information to split the new version of the total NMVOC emissions from the EDGAR inventory (Janssens-Maenhout et al., 2016) into individual species, which are then lumped to 25 species groups (Olivier et al., 1996) as proposed within the Global Emission Inventory Activity (GEIA). Finally, global grid maps from 1970 to 2010 are developed at $0.1° \times 0.1°$ resolution for different sectors and species groups.

### 2.1 Revised total NMVOC emissions of EDGAR

The NMVOC emissions described in this work refer to the EDGARv4.3.2 dataset which includes annual emissions for 228 countries from the year 1970 until 2012. All anthropogenic activities have been grouped into 14 main emission sectors: power generation (IPCC_1996 categories 1A and 1B), combustion for manufacturing industry (IPCC_1996 code 1A2), energy for buildings (IPCC_1996 code 1A4), road transportation including evaporative emissions for gasoline related fuels (IPCC_1996 code 1A3b), transformation industry (IPCC_1996 codes 1A1c+1A5b1+1B1b+1B2a6+1B2b5+2C1b), fugitive emissions from fuel exploitation (IPCC_1996 codes 1B1a+1B2a1+1B2a2+1B2a3+1B2a4+1B2c), process emissions during production and application including production of chemicals, paper/food/iron and steel production/ solvent use (IPCC_1996 codes 2+3), oil refineries (IPCC_1996 codes 1A1b+1B2a5), agricultural waste burning (IPCC_1996 code 4F), shipping including both domestic and international shipping (IPCC_1996 codes 1A3d+1C2), railways, pipelines and off-road transport (IPCC_1996 code 1A3c+1A3e), fossil fuel fires (IPCC_1996 code 7A), solid waste and wastewater (IPCC_1996 code 6), aviation differentiating among climbing and descent, cruise, landing and take-off and supersonic (IPCC_1996 code 1A3a). No large scale biomass burning emissions are estimated in the current work.

EDGAR activity data were mainly retrieved from the IEA energy statistics (IEA, 2014) for the fuel consumption, from Commodity Statistics of UN STATS (2014) and USGS (2014) for production processes and from FAO (FAO STAT, 2014)



for agriculture. Further details on activity data by sector can be found in Janssens-Maenhout et al., (2016). NMVOC emission factors are consistent with the EMEP/EEA 2013 Guidebook (EEA, 2013), while abatement measures are implemented for the road transport sector (consistent with the Euro standards), for the production of chemicals (CHa-formaldehyde (methanal), total polyethylene, CHa-propylene glycol, total polystyrene), for power generation (auto

produced electricity and public electricity production from natural gas) and for landfills. Figure S1 of the supplementary material shows the comparison of global NMVOC emissions by sector for different EDGAR versions v4.2 (refer to http://edgar.jrc.ec.europa.eu/overview.php?v=42), v4.3.1 (refer to http://edgar.jrc.ec.europa.eu/overview.php?v=431) and v4.3.2 (http://edgar.jrc.ec.europa.eu/overview.php?v=432_VOC_spec&SECURE=123) for the most recent year (2008) available for all datasets. Total emissions are slightly higher (ca 17%) in the current version of EDGAR compared to v4.3.1

mainly due to changes in the activity data and emission factors. At sector level, rather good agreement is observed between EDGARv4.3.2 and EDGARv4.3.1, although major differences are found for the application of solvents showing 15.6 times higher emissions for EDGARv4.3.2 due to revised activity data (to account for household products and other solvents use) and emission factors (especially for paints and pesticides), the residential and transformation industry sectors having ca 30% and 22% lower emissions. Finally, in EDGARv4.3.2 waste water treatment and glass production (from the year 1990) have

been introduced.

Figures S2 and S3 show the comparison of NMVOC emissions of EDGARv4.3.2 and the best estimates provided by the HTAP_v2.2 inventory for the year 2010 by HTAP sector and country (refer to Janssens-Maenhout et al. (2015) and http://edgar.jrc.ec.europa.eu/htap_v2/index.php). Very good agreement for all sectors is found between EDGARv4.3.2 and HTAP_v2.2 for Asian countries and North America (refer to Fig. S2), as well as for Europe (refer to Fig. S3). Significant

underestimation of EDGARv4.3.2 NMVOC emissions is observed for India and Indonesia for the residential and transport sectors (although the reported HTAP_v2.2 emissions appear to be very high compared for example with the Chinese ones). On the other hand, EDGARv4.3.2 significantly overestimates German NMVOC emissions for the residential sector, although the HTAP_v2.2 data appear to be too low compared for example with France residential emissions. In general, larger differences between the two inventories are observed for the power generation due to the low NMVOC emissions

associated with this sector.

Focusing on European countries (see Fig. 1), detailed comparison by sector and country (defined with ISO codes) is also performed with officially reported EEA NMVOC emission inventories for the year 2010 (http://www.eea.europa.eu/data-and-maps/data/national-emissions-reported-to-the-convention-on-long-range-transboundary-air-pollution-lrtap-convention-10). Total NMVOC emissions at European scale are overestimated by 15% by EDGAR compared to EEA and HTAP_v2.2.

However, insights on the origin of such differences can be retrieved looking at sectorial emissions. The power generation sector in EU represents less than 2% of total NMVOC emissions and it is characterized by the largest discrepancies among inventories. Overall, the EDGARv4.3.2 power emissions are 60% lower compared to the HTAP_v2.2 data, while they are 60% higher compared to the EEA official data. Just to give some examples, German emissions are underestimated by EDGAR by 75% compared to HTAP, while they are overestimated by 50% compared to EEA. On the other hand, UK





emissions are overestimated by a factor of 6 and 9 by EDGAR and HTAP, respectively, compared to the reported EEA data. As shown in Fig. 1 and Fig. S3, industrial, residential and ground transport NMVOC emissions are characterized by better agreement among the three inventories, with the exception of few countries. EDGAR underestimates by 30-50% ground transport emissions of France, Poland and Czech Republic compared to HTAP and EEA, while it generally overestimates
residential emissions (e.g. in particular for Germany, France and UK).

## 2.2 Data sources for NMVOC speciation profiles

A review of the available literature and databases applicable to NMVOC speciation profiles in different regions was undertaken. Theloke and Friedrich (2007) provide a database (IER database) of 87 speciation profiles for Europe. The profiles distribute total NMVOC emissions of anthropogenic NMVOC sources into 305 single NMVOC species or species
classes. The IER database is widely used for emission analysis and creating input data for atmospheric dispersion models in Europe (Coll et al., 2010; Kühlwein et al., 2002; Vautard, 2003). This database is used as the main data source for the profiles mapping of Europe. The joint EMEP/EEA air pollutant emission inventory guidebook (EMEP/EEA, 2014) provides very detailed documentation on a sectorial basis for a number of pollutants. The NMVOC speciation profiles for the road transport sector were extracted from the EMEP/EEA guidebook.
In the absence of a comprehensive and elaborated NMVOC profiles database for Asia, NMVOC speciation profiles from local studies have been systematically collected and analysed for different sources, including solvent use (Lau et al., 2010; Wang, 2014; Yuan et al., 2010), transport (Cai and Xie, 2009; Fu, 2008; Lau et al., 2010; Lu, 2003; Wei et al., 2012), fuel burning (Cai et al., 2010; Lau et al., 2010; Liu et al., 2008; Wei et al., 2012), biomass burning (Cai et al., 2010; Li et al., 2009; Wei et al., 2008), petrochemical industry (Lau et al., 2010; Liu et al., 2008; Wei et al., 2012), coking (He et al., 2005;
Jia et al., 2009; Wei et al., 2012), production and manufacturing industry (Cai et al., 2010; He et al., 2012; Klimont et al., 2002), and waste disposal (Klimont et al., 2002). Information on NMVOC speciation in Asia was generally available only for China or for single Chinese regions, e.g. Shanghai (Cai et al., 2010) or Pearl River Delta (Chan et al., 2006). Given that China is the biggest NMVOC emitter in Asia and the lack of data for other Asian countries, it is assumed that the speciation profiles collected for China are representative for Asia.
In North America, reference material is typically well co-ordinated and centralised at national level by the United States Environmental Protection Agency (US EPA). The SPECIATE4.4 data set (Hsu et al., 2014), hereafter referred to as SPECIATE, is the most comprehensive data set available for North America, containing 1879 unique speciation profiles for VOC emissions disaggregated to 1717 individual species from an extensive list of sources. NMVOC speciation profiles for North America were extracted from the SPECIATE database.
Local studies or measurements of NMVOC composition of emission sources for other regions (e.g. Africa, Latin America) are too limited to support the generation of local NMVOC speciation database. Given that the SPECIATE database is the most comprehensive data source, and is already widely used in locations where local data are not available, in our study we used it as the data source of NMVOC profiles for these regions.





When screening NMVOC speciation profiles, we have given preference to existing databases built on large amount of studies. Researche and articles that have already been taken into account in the above mentioned review paper and databases are not listed here.

## 2.3 Compilation and mapping of speciation profiles

Total NMVOC emissions from the EDGAR v4.3.2 emission inventory (http://edgar.jrc.ec.europa.eu/overview.php?v=432_VOC_spec&SECURE=123) are speciated at country- and process-disaggregated levels.

NMVOC speciation profiles collected from the different databases and publications were mapped to all EDGAR process codes which were retrieved from the EDGAR v4.3.2 emission inventory. The first step in this approach was to match the

EDGAR process codes for which there was an exact or similar match in the corresponding profiles database, i.e. a match between both the source and fuel type (as shown in Table S2). If an exact match was unavailable, fuel-specific speciation profiles were assigned to EDGAR process codes (Table S3 and Table S4). This process was continued, assigning the best available matches in the profiles data set to EDGAR processes (Table S5). In many cases this involved expert judgement determining the best available profile where no sector or fuel-specific profiles were present in the speciation profiles data set.

However, the expert judgement was guided by a detailed knowledge of the emission characteristics of different sources, allowing the best available matches to be made, and providing a complete, gap-filled data set.

For processes where similar technologies are used in different regions (e.g. boilers, vehicles) and local profiles are not available, profiles from databases of other regions (e.g. SPECIATE database, IER database) were used for filling the gaps within the data set for a certain region (e.g. China). As for regions other than Europe, Asia, and North America, the mapping

made using the SPECIATE database is suggested to be taken as a general estimation of source oriented NMVOC composition.

Most of the available speciation profiles are fuel-oriented, e.g. for "coal combustion" processes, and do not always match the scope of the sector activities, e.g. "energy industry". In order to show how well the assigned NMVOC profile matches the corresponding EDGAR process, codes indicating the level of the matching quality were assigned to each mapping (see Table

1). Six levels of mapping quality codes are defined, which not only indicate how specific a match is, but also imply priorities of further improvement. A quality code of 1 to 4 is considered to be a relatively good match and representative of the EDGAR process. Quality codes 5 and 6 represent fuzzy matches due to the lack of process-specific profiles and are considered to be the priority areas for further improvement.

Table S2 presents examples of chemical processes for which an exact match was assigned with the profiles database. The

assigned profiles are specific to EDGAR source codes and there is no differentiation in technology codes within these EDGAR processes. Table S3 presents an example of a profile that is considered to be representative of the EDGAR process but not an exact match; for example the profile of external combustion boiler is identified as the best available match to the public cogeneration process. In this case, a mapping quality code of 2 is assigned. Table S4 shows an example, where no





differentiation of profiles for technology codes was possible. In this case, only biodiesel profiles for light duty trucks are available in the profiles database and have been applied to processes for buses and heavy duty vehicles. These mappings are assigned with quality code 3. Table S5 shows examples of profiles that are considered to be fuel-specific only (quality code 4), a general profile (quality code 5) and a fuzzy match (quality code 6), respectively.

**2.4 Species aggregation**

In order to integrate the NMVOC speciation data from different databases into EDGAR and to provide a data set fitting the computational requirements of chemistry models, individual NMVOC species in speciation profiles are aggregated to groups. A review and consultation with modellers and experts regarding NMVOC species aggregation mechanisms were conducted. It was decided to aggregate single NMVOC species to the 25 species groups proposed within GEIA, as detailed
in Table 2 where a general molecular formula and the photochemical ozone creation potential (POCP) are also provided. POCP values were calculated through weighted averages of the values reported in Dore et al. (2006) for the 50 most significant NMVOC species reported in the UK. Given the limited data available, the assigned POCP values are considered to be an estimate and not an accurate representation of the GEIA groups.

Lists of all the unique species present in different databases (i.e. SPECIATE, IER database) were created. Each species was
15 then assigned to one of the GEIA 25 species groups. The general NMVOC species grouping methodology suggested by Carter (2015) was taken. Where a species contains more than one functional group, priority was typically given to the suffix of the species name since this functional group is generally the most relevant for ozone formation.

**2.5 Development of grid maps**

The EDGAR v4.3.2 speciated NMVOC emissions are available both as time series by sector and country (1970-2012) and as
global grid maps by sector at the following website http://edgar.jrc.ec.europa.eu/overview.php?v=432_VOC_spec&SECURE=123. Gridmaps are available every 10 years from 1970 to 2000 and with annual resolution from 2000 to 2012. The analysis of NMVOC emission time series and 2010 speciated grid maps is presented in section 3.1.

**3 Results**

A global data set providing information about NMVOC composition in the level of 25 NMVOC groups of each EDGAR process from 1970 to 2012 was developed and integrated into the EDGAR database. The compiled NMVOC speciation profiles and their allocation to EDGAR processes and IPCC sectors are available as supplementary data to this article. The total NMVOC emissions in the EDGAR database were then disaggregated with speciation data.



### 3.1 Species time series 1970-2012 and 2010 grid maps

Over the past four decades, global NMVOC emissions increased from 119000 to 169000 ktons, although different regional trends can be observed, as shown in Fig. 3. North America and Europe halved their emissions from 1970 to 2012, while Africa, China, India and the rest of Asia increased their emissions by factors of 2.9, 2.5, 2.2 and 1.8, respectively. Nowadays,

top emitter countries are Asia and Africa producing ca 65% of global NMVOC, while North America and Europe contribute only to 14% (in comparison to 1970 when they contributed 37% of global NMVOC emissions). The reduction in American and European NMVOC emissions has happened mainly in the road transport sector (affecting both evaporative and combustion emissions) and residential combustion due to the implementation of reduction measures (Euro standards) combined with the use of cleaner fuels. Global NMVOC emissions are mainly produced by road transportation, residential

combustion, transformation industry, fuel production and transmission and solvent use, representing 16%, 15%, 18%, 16% and 12% of 2010 total NMVOC emissions, respectively.

Figure 4 represents an example of global grid maps obtained for a single NMVOC species of the EDGARv4.3.2 dataset. As reported in Fig. 4, in 2010 we observe a significant reduction in methanal emissions, in particular over Europe, which can be attributed to the adoption of increasingly stringent Euro standards compared to the year 2000. A similar pattern is also

observed for benzene emitted by the same sector.

Figure 5 and Figure 6 represent 2010 total NMVOC gridmaps for the residential and road transport sectors. In addition, the relative contribution of the NMVOC species to each sector is reported in the pie charts for major world regions. The highest NMVOC emissions for the residential sector are observed in Africa (7.9 ktons), China (5.2 ktons) and India (4.3 ktons) in 2010. In 2010 60% of global NMVOC emissions from the residential sector are attributed to aldehydes (grouped here as

alkanals) mainly emitted from biomass combustion (refer to Fig. S4a); however, a different composition of NMVOC emissions is retrieved for different world regions, as shown in Fig. 5, USA, Latin America and Africa are characterized by a rather similar composition of residential emissions (alkanals, aromatics and "other VOCs") partly reflecting the gapfilling procedure using the SPECIATE database. In addition to aromatics (alkanones, dimethylbenzenes and benzene) and alkanals (aldehydes), EU residential emissions are characterized by alkenes (ethane) and alkynes. Chinese and Indian residential

emissions are dominated by alkenes (ethane), alk(adi)enes/alkynes (ethyne and olefines) and other VOCs (e.g. ketones).

Similarly to Fig. 5, Fig. 6 represents total and speciated NMVOC emissions for the road transport sector in 2010, including both combustion and evaporative emissions. Latin America (5.1 ktons), USA (3.4 ktons) and China (3.0 ktons) are the top emitters for this sector, while Europe (0.8 ktons) is the lowest emitter due to the higher share of diesel vehicles compared to petrol engine vehicles (which are also subjected to evaporative emissions, see Fig. S4b). Overall, road transport emissions

are dominated by C2-C5 and C6+ alkanes and aromatics (e.g. toluene); in addition contributions from alk(adi)enes/alkynes (olefins) and alkenes (ethane) are observed for Europe and USA. Latin America is strongly dominated by alkanoic acids due to the higher share of biofuel used for road transport (refer to Fig. S4b).



## 3.2 Case study on the impact of reduction measures on speciated NMVOC emissions

### 3.2.1 Case study of Germany

The resulting data set permits on analysis of the trends of speciated NMVOC emissions by source and region, e.g. for road transport and residential sectors of Germany. The total NMVOC emissions from road transport in Germany decreased

steadily since 1990s as shown in Fig. 7. The percentages of alkanes (propane, butanes, and pentanes) increased consecutively especially in recent years, while the proportion of aromatics decreased. These trends reflect the impacts of transport emission control strategies and the utilization of cleaner fuels in Germany. According to the EMEP/EEA air pollutant emission inventory guidebook for road transport (EMEP/EEA, 2014), NMVOC emissions from closed-loop-catalyst (Euro 1 and later) gasoline four stroke vehicles have a higher composition of alkanes, and lower content of aromatics compared with that of

conventional gasoline vehicles. The number of LPG vehicles in Germany increased from 40,000 in 2006 to around 370,000 in 2010 (KBA, 2010). The contents of alkanes (aromatics) of emissions from LPG vehicles are much higher (lower) than that from gasoline or diesel vehicles when comparing the corresponding profiles.

Residential NMVOC emissions in Germany decreased by more than 80% from 1986 to 2000, and then became relatively steady in recent years (Fig. 8). The composition changes were however more substantial then occurred in the road transport

sector. The percentages of alkenes and alkanals emissions increased, whilst the proportion of alkanes (C6+) and aromatics decreased over the considered time frame. These changes are related to the fuel shift from peat and coal to oil, gas, and solid biomass in the residential sector in Germany. Figure S5 presents the contribution of different types of fuel to NMVOC emissions of residential sector in Germany from 1970 to 2012. NMVOC emissions from combustion of peat accounted for over 50% of residential NMVOC emissions in Germany in 1980s, and decreased to 8% in 2010. Meanwhile the percentages

of oil, gas and biofuel (mainly primary solid biomass) related NMVOC emissions increased from 6% in 1970 to over 80% in 2010. NMVOC emissions from combustion of peat have higher contents of alkanes (C6+) and aromatics (dimethylbenzenes), and lower contents of alkenes and alkanals, compared with those of oil, natural gas and solid biomass combustion (Theloke and Friedrich, 2007).

With speciated NMVOC emissions data, the impacts of emission control policies by certain regions on the emission amount

of specific NMVOC species as well as the NMVOC composition changes could be investigated and analysed.

### 3.2.2 Case study of the United Kingdom

The resulting dataset for the UK is comparable with the UK's National Atmospheric Emission Inventory (NAEI). There are slight differences in the absolute values as this study was based on international data, due to a lack of available national data for this study. However, the overall trends are the same. Similar to the trends in Germany (see Section 3.2.1), NMVOC

emissions began to decline in 1990 for all major sources and decreased well below the Gothenburg Protocol Ceiling in 2010. Emissions in the UK have reduced by approximately 70% between 1990 and 2010. This reduction has been driven by a



number of key factors and legislation including the Directive on Industrial Emissions (2010/75/EU), the Solvents Directive (99/13/EC) and the Convention on Long Range Transboundary Air Pollution (CLRTAP).

Specifically, for road transport emissions, the reduction in NMVOC emissions has been driven by the requirement for all new petrol cars to be fitted with three-way catalysts since 1989 and by fuel switching from petrol to diesel (UK IIR, 2015).

Approximately 90% of NMVOC emissions from road transport attributed to petrol vehicles. UK NMVOC emissions are presented in Fig. 9 and Fig. 10.

# 4 Quality assessment and data comparison

## 4.1 Quality assessment by region

The availability of NMVOC speciation profiles varies among different sources. Whilst some detailed data are available for

selected sources, the speciation profiles are not necessarily accompanied by information on either the accuracy or the general quality of the data. In addition, it is challenging to quantify the impact on uncertainty of some of the data handling steps (e.g. expert judgement in allocation of speciation profiles to particular EDGAR processes, and the allocation of species to species groups). It was therefore considered that a qualitative approach was the most appropriate method for expressing uncertainties, by using "quality codes".

The use of "quality codes" is an approach commonly used (e.g. by the US EPA) when uncertainties are particularly large and/or difficult to quantify. Comparison of the types of profile matches that were made in this study using different databases (i.e. SPECIATE and IER data sets) led to the formation of six levels of quality codes (see Table 1). This was considered to be the best method to enable the quality of the assigned speciation profiles to be recognised, and to give a clear indication of the quality of the match between the EDGAR process and the assigned speciation profile.

Figure 11 provides a summary of the percentages of NMVOC emissions associated to each of the quality codes for Europe, China and North America. Two years of data (2010 and 2000) are presented to reduce the bias caused by choosing a specific year. 42% (2010) and 55% (2000) of NMVOC emissions in Europe are attributable to the sources to which speciation profiles with quality code 1 (well matched) are mapped. For the emissions in China, 81% and 76% of NMVOC emissions are associated with fuel-specific speciation profiles that are not sector-specific, i.e. quality code 4, owing to poorer data

availability in China. 44% (2010) and 48% (2000) of NMVOC emissions in North America are generated by the sources mapped with profiles of quality code 3 (sector- and fuel-specific). Percentages of quality 5 and 6 related with NMVOC emissions in the three regions in both years are less than 13%.

## 4.2 Quality assessment by source

The results are further disaggregated to different source groups as defined in the EDGAR database. Figure 12, Figure 13 and

30 Figure 14 display the NMVOC emissions associated with each quality code of the 17 EDGAR source groups in Europe, China, and North America respectively in 2010. Road transport and the residential sectors are the largest NMVOC emission





sources in Europe in 2010 according to the EDGAR v4.3.2 emission data. Road transport processes mapped with well-matched (quality code 1) speciation profiles contribute to 96% of total NMVOC emissions of this source. This is primarily due to the good data quality of NMVOC speciation profiles extracted from the EMEP/EEA Guidebook (EMEP/EEA, 2014). Up-to-date speciation profiles are provided for different vehicle types (e.g. light or heavy duty vehicles, passenger cars, etc.),

fuel types (e.g. gasoline, diesel, etc.), and end-of-pipe technologies (e.g. pre and after Euro I standards). Chemicals production and solid waste disposal are the NMVOC emission sources for which high quality speciation profiles are not available. However their contributions to total NMVOC emissions are much smaller than road transport and residential sources (as shown in Fig. 12).

As a result of poorer data availability, the profiles mapping quality for China is considered lower than that of Europe.

Residential, fuel production, and manufacturing industry are the major NMVOC emission contributors in China in 2010 (see Fig. 13). Most of the NMVOC emissions of these three sources are mapped with fuel-specific speciation profiles (e.g. coal combustion, oil combustion). Solid waste disposal and production of pulp and paper are the sources lacking of high quality speciation profiles and with relatively high contribution to total NMVOC emissions.

In North America, road transport and fuel production sources contribute to 56% of total NMVOC emissions in 2010 (see

Fig. 14). The speciation profiles from SPECIATE database for road transport are mostly sector- and fuel-specific, but not always with vehicle type specification (categorized to quality code 3). These profiles are related to 87% of road transport NMVOC emissions. 74% of NMVOC emissions from fuel production sources are mapped with fuel-specific speciation profiles (quality code 4). 90% of NMVOC emissions from non-road transport sources are mapped with speciation profiles with a quality code of 6, resulting from the lack of profiles for shipping in the SPECIATE database. Good quality speciation

profiles for food production sources are also not available. However given its small contribution (0.8%) to total NMVOC emissions, the related impact to data quality is believed to be negligible.

## 4.3 Quality assessment by species group

Figure 15, Figure 16 and Figure 17 show the emissions of 25 NMVOC species groups associated to each quality code in 2010 in Europe, China, and North America. In Europe, hexanes and higher alkanes, other NMVOC, and ethene are the

dominant NMVOC species groups, which contribute to 10.4%, 9.1%, and 8.2% of total NMVOC emissions in 2010, respectively (see Fig. 15). 45% of emissions of hexanes and higher alkanes are generated from processes mapped with well-matched speciation profiles (quality code 1). Emissions of ethene from processes mapped with fuel-specific speciation profiles contribute to 60% of total ethene emissions. Emissions from processes with quality 5 and 6 profiles mappings account for less than 23% for all species groups except esters.

As can be seen in Fig. 16, chlorinated hydrocarbons, hexanes and higher alkanes, and ethene are the most abundant NMVOC species groups in China in 2010. Profiles mappings classified as fuel-specific (quality code 4) are related to over 45% of total emissions of almost all the 25 species groups except for isoprenes, trimethylbenzenes, and acids, which only account for in total 0.3% of total NMVOC emissions in China. In North America (Fig. 17), hexanes and higher alkanes, other


NMVOC, and butanes are the most important NMVOC species groups, which contribute to 16.6%, 10.7%, and 8.3% of total NMVOC emissions respectively in 2010. Emissions of hexanes and higher alkanes from processes with sector and fuel-specific (quality code 3) profiles mapping account for 60% of total hexanes and higher alkanes emissions. 50% of emissions of other NMVOC are related to processes mapped with well-matched (quality code 1) speciation profiles. For NMVOC

species groups except alkanols, butanes, and pentanes, emissions with quality 5 and 6 profiles mappings account for less than 8% of total emissions.

### 4.4 Data comparison

The EDGAR NMVOC speciation is compared with the RETRO (REanalysis of the TROposhperic chemical composition) emission inventory (Schultz et al., 2007), in which speciated NMVOC emission data are provided, although in a much

coarser sectoral structure.

The RETRO emission inventory provides global gridded data sets for anthropogenic emissions covering the period from 1960 to 2000. NMVOC emissions are disaggregated to 25 NMVOC species groups and eight sectors. Each species group of NMVOC emissions of all sectors for Europe, China and the United States (US) is extracted from the RETRO emission grid maps. A matching between RETRO sectors and EDGAR sources is made (as shown in Table S6) based on sector definitions,

to prepare data from two inventories ready for comparison.

Figure 18 presents the comparison of NMVOC species composition of eight sectors between EDGAR and RETRO data sets for Europe, China and the United States for the year 2000. It can be seen that the RETRO emission inventory shows generally the same or similar NMVOC composition across Europe (R_EU), China (R_CN), and the United States (R_US) for all the eight sectors, while the speciated EDGAR NMVOC emission data (E_EU, E_CN and E_US) gives a different

picture on this issue. Regional variations of NMVOC composition could be seen across the investigated sectors from EDGAR emission data, which reflect the results of different fuel structure, technologies and legislations among the three regions. Characteristics of NMVOC emissions composition of each sector in different regions could be better identified and analysed with the data sets produced in this study.

Generally the RETRO data set agrees well with the EDGAR European data set, especially for fuel extraction, residential and

25 transport sectors which are the main sources of NMVOC emissions. It indicates that the RETRO data set may have been used mainly for European NMVOC profiles. For China, high contents of chlorinated hydrocarbons (classified as other NMVOCs in Fig. 18) emissions from fuel extraction, industrial and power generation sectors are recognisable from the EDGAR data set. Coal combustion is the main source of NMVOC emissions in those sectors in China. NMVOC speciation profiles of coal combustion source collected from local studies (Cai et al., 2010; Liu et al., 2008; Wei et al., 2012) support

this characteristic reflected by the EDGAR data set. For the United States, the EDGAR and RETRO data sets agree well for the NMVOC composition of industrial and power generation emissions. Solvent use and Transport sectors also have similar pictures. These four sectors contributed to over 50% of total NMVOC emissions in the United States in 2010 according to the EDGAR database.



# 5 Conclusions and outlook

In this study, a global speciated NMVOC emission data set is developed by compiling and allocating region- and source-specific NMVOC speciation profiles to the EDGAR v4.3.2 emission inventory, which can serve as input data for CTMs and related health impact assessments. This, to the authors' knowledge, represents the first compilation of global speciated NMVOC grid maps with high resolution. Quality codes are assigned to each matching to indicate the appropriateness and completeness. Individual NMVOC species in different profiles are aggregated to 25 species groups as proposed by the GEIA initiative to enhance the usability of the data set for CTMs. By integrating the speciation profiles into the EDGAR database, species time series for the period 1970-2012 and global grid maps of $0.1° \times 0.1°$ are produced by sector and species. Trends of NMVOC emissions from 1970 to 2012 are analysed by region. Case studies for Germany and the UK show that total NMVOC emissions of transport and residential sectors decreased dramatically from late 1980s to recent years in both countries. Implementation of transport emission control strategies and the fuel shift from coal to cleaner fuels (oil, natural gas, and solid biomass) have led to increased shares of alkanes and alkanals, and decreased share of aromatics. A quality assessment is performed to discuss the uncertainties of the generated data set by region, source, and species. Comparison of the generated emission data set with the RETRO emission inventory shows good agreement for sectors in Europe and the United States, and higher regional specificity of the produced data set.

Due to the unavailability of measurements and literature of local NMVOC source profiles, regional specific profiles for regions other than Europe, Asia, and North America (e.g. Africa, South America) could not be compiled. Instead, a 'world average' profile set has been generated, that is recommended for use in these regions. Six levels of mapping quality codes are assigned when speciation profiles are mapped to certain EDGAR process, with quality code 5 and 6 representing fuzzy matches due to the lack of specific profiles. These represent the priority areas for further improvement.

## Acknowledgements

This work was financed under the project "Provision of Support for Atmospheric composition modelling-Case Study-Mapping of NMVOC Emissions in the EDGAR system" by the Joint Research Center, European Commission. We would like to thank Erika von Schneidemesser of Institute for Advanced Sustainability Studies and Frank Dentener of European Commission, Joint Research Centre, Directorate for Sustainable Resources, Food Security Unit for their insights and suggestions on species aggregation.

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





**Table 1. The quality codes used to describe the quality of the match between the speciation profile and the EDGAR process.**

| Quality code | Description |
| --- | --- |
| 1 | Well matched. |
| 2 | Well matched, fuel differentiation not fully addressed (e.g. biogasoline vs gasoline). |
| 3 | Sector-, fuel-specific; technology not differentiated (e.g. transport data that is not specific to a vehicle type). |
| 4 | Fuel-specific; sector, technology not differentiated. |
| 5 | Catch all processes, a general profile that provides a best available match. |
| 6 | Best profile available, not considered to be a specific match. |



**Table 2. List of GEIA 25 NMVOC groups with molecular formulae and Photochemical Ozone Creation Potential (POCP).**

| GEIA ID | GEIA group | Molecular formula | POCP (Derwent et al., 1998) |
|---|---|---|---|
| voc1 | Alkanols (alcohols) | $C_nH_{2n+1}OH$ | 34.92 |
| voc2 | Ethane | $C_2H_6$ | 12.30 |
| voc3 | Propane | $C_3H_8$ | 22.12 |
| voc4 | Butanes | $C_4H_{10}$ | 36.54 |
| voc5 | Pentanes | $C_5H_{12}$ | 39.50 |
| voc6 | Hexanes and higher alkanes | $C_nH_{2n+2}$ ($n \geq 6$) | 44.15 |
| voc7 | Ethene (ethylene) | $C_2H_4$ | 100.00 |
| voc8 | Propene | $C_3H_6$ | 97.89 |
| voc9 | Ethyne (acetylene) | $C_2H_2$ | 8.50 |
| voc10 | Isoprenes | $C_5H_8$ | 109.20 |
| voc11 | Monoterpenes | $C_{10}H_{16}$ | 109.20* |
| voc12 | Other alk(adi)enes/alkynes (olefines) | $C_nH_{2n-2}$ | 95.29 |
| voc13 | Benzene (benzol) | $C_6H_6$ | 21.80 |
| voc14 | Methylbenzene (toluene) | $C_7H_8$ | 63.70 |
| voc15 | Dimethylbenzenes (xylenes) | $C_6H_4(CH_3)_2$ | 107.41 |
| voc16 | Trimethylbenzenes | $C_6H_3(CH_3)_3$ | 129.86 |
| voc17 | Other aromatics | $C_nH_{2n-6}$ | 77.78 |
| voc18 | Esters | R-C(=O)O-R' | 20.68 |
| voc19 | Ethers (alkoxy alkanes) | R-O-R' | 12.44** |
| voc20 | Chlorinated hydrocarbons | $CH_3Cl$ | 23.72 |
| voc21 | Methanal (formaldehyde) | $CH_2O$ | 51.90 |
| voc22 | Other alkanals (aldehyedes) | R-CHO | 64.10 |
| voc23 | Alkanones (ketones) | R-C(=O)-R' | 24.54 |
| voc24 | Acids (alkanoic) | $R-C_nH_nCOOH$ | 12.44** |
| voc25 | Other NMVOC (HCFCs, nitriles, etc.) | NA | 12.44 |

\* Value was not available, has been assumed to be the same as "isoprenes"

\*\* Values not available, assigned the same value as "other"

Notes: R and R' denote functional groups. Where general formulae are not appropriate, the simplest molecular formula representing the group is provided. NA = not available



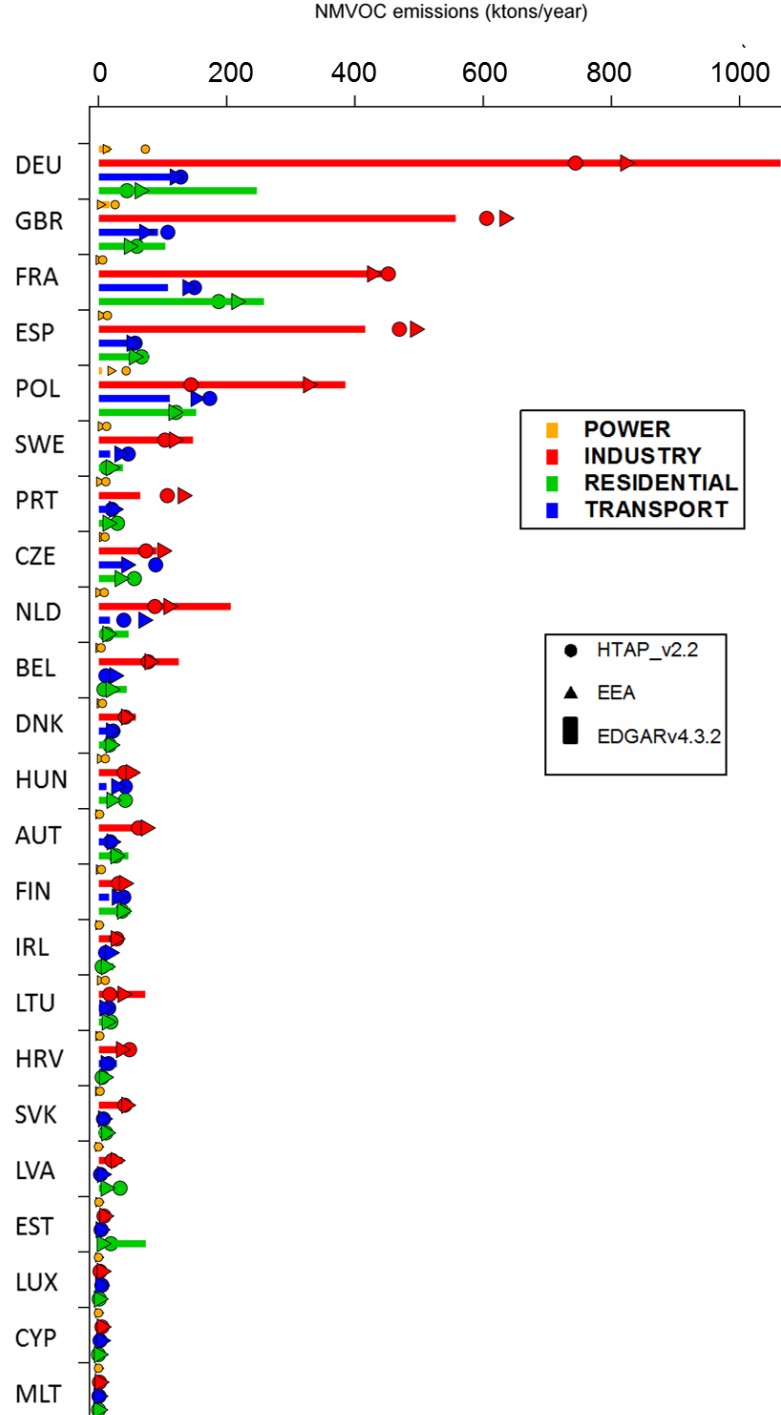

**Figure 1.** Comparison of 2010 NMVOC emissions from the power generation, industry, residential and combustion sectors of the HTAP_v2.2, EDGARv4.3.2 and EEA inventories.





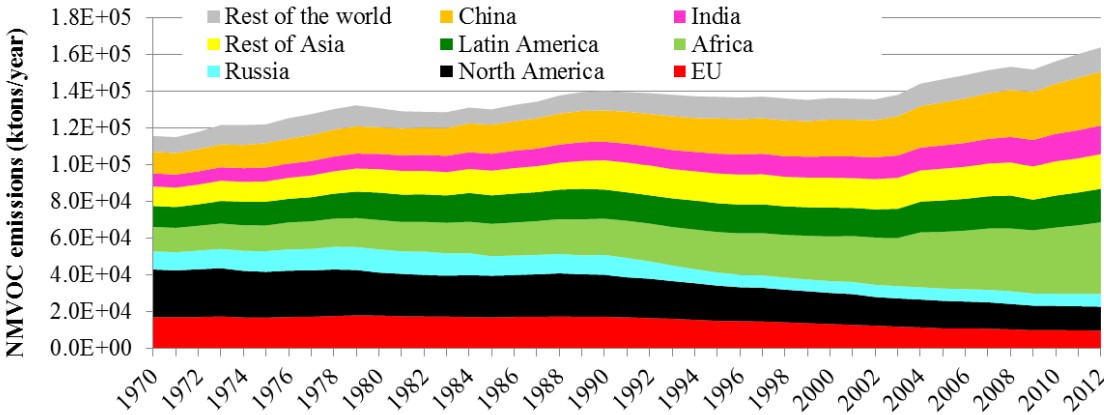

**Figure 2. Global trend of NMVOC emissions by region.**







**Figure 3. Comparison of 2000 and 2010 methanal emission gridmaps from the road transport sector.**



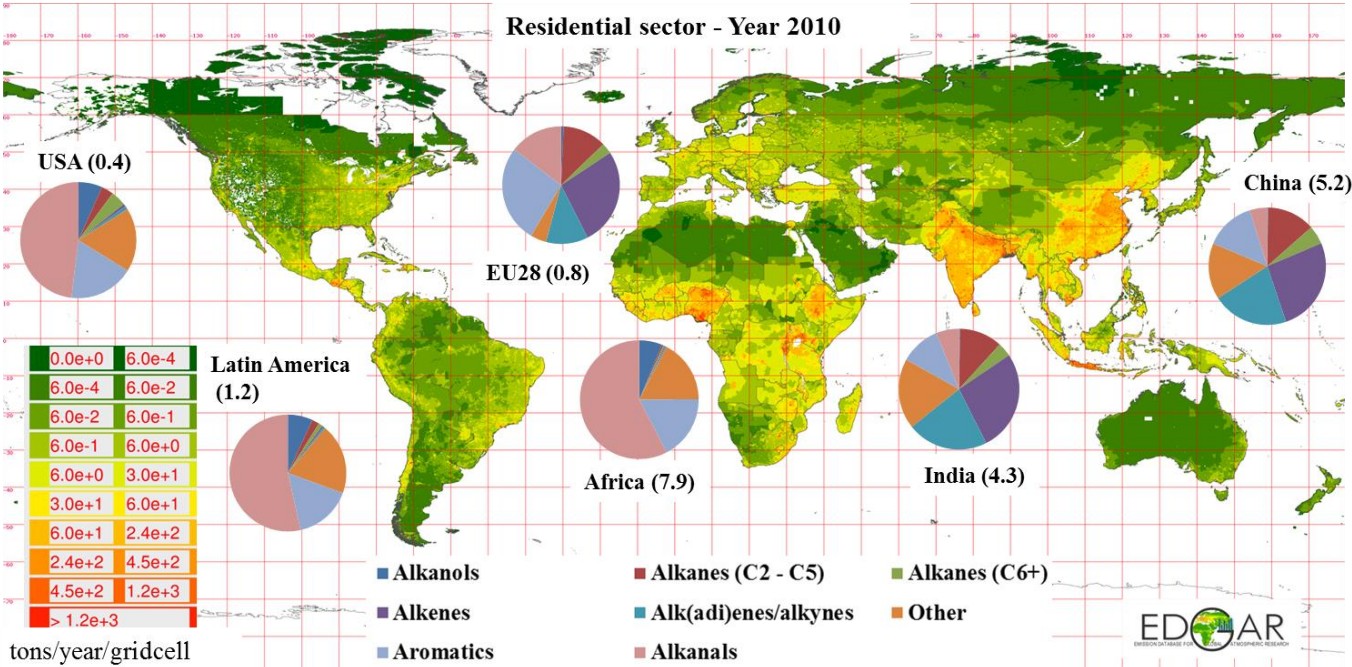

**Figure 4. NMVOC emission gridmap from the residential sector in 2010. The relative contribution of NMVOC species is reported in the pie charts for major world regions (number in brackets refer to total NMVOC emissions (in ktons) for the residential sector for each region).**





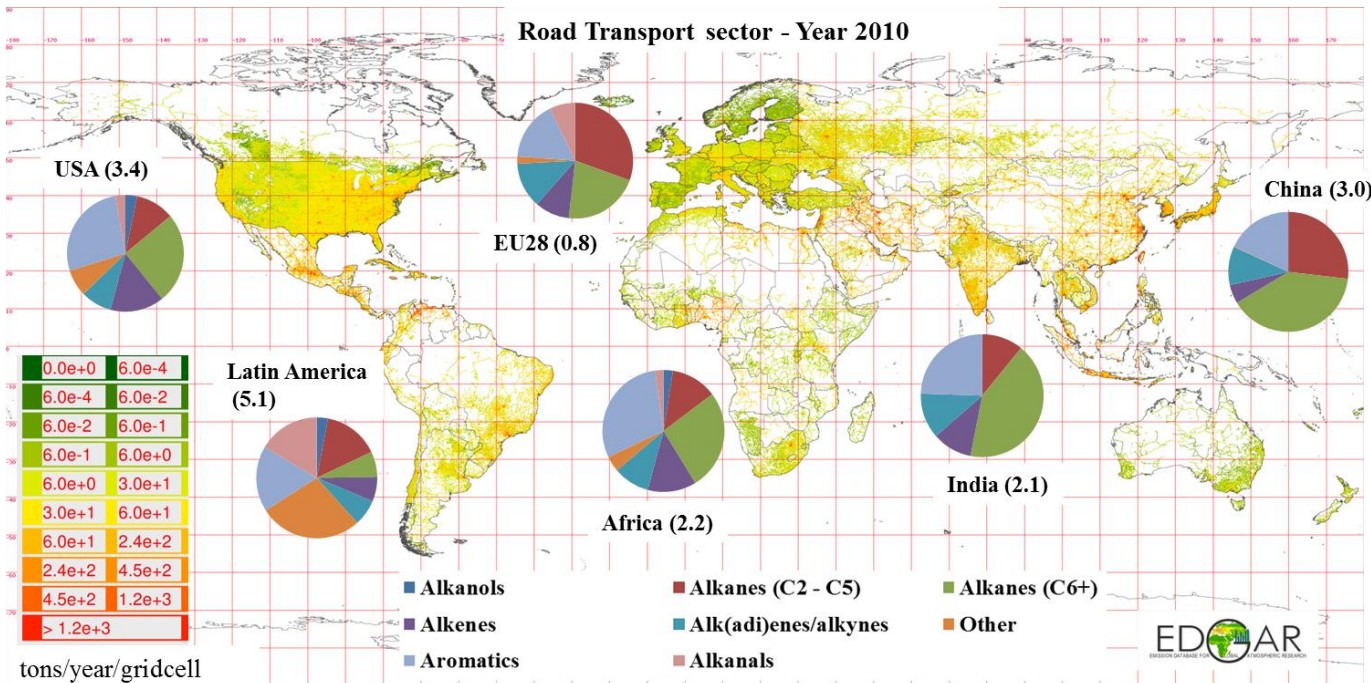

**Figure 5. NMVOC emission gridmap from the road transport sector in 2010. The relative contribution of NMVOC species is reported in the pie charts for major world regions (number in brackets refer to total NMVOC emissions (in ktons) for the road transport sector for each region).**




**Figure 6. Total NMVOC emissions and their speciation for the road transport sector in Germany during 1970-2012.**



**Figure 7. Total NMVOC emissions and their speciation for the residential sector in Germany during 1970-2012.**





**Figure 8. Total NMVOC combustion emissions and their speciation for petrol vehicles of the road transport sector in the UK during 1970-2012.**



**Figure 9. Total NMVOC combustion emissions and their speciation for diesel vehicles of the road transport sector in the UK during 1970-2012.**



**Figure 10. Percentages of NMVOC emissions associated to each quality code.**



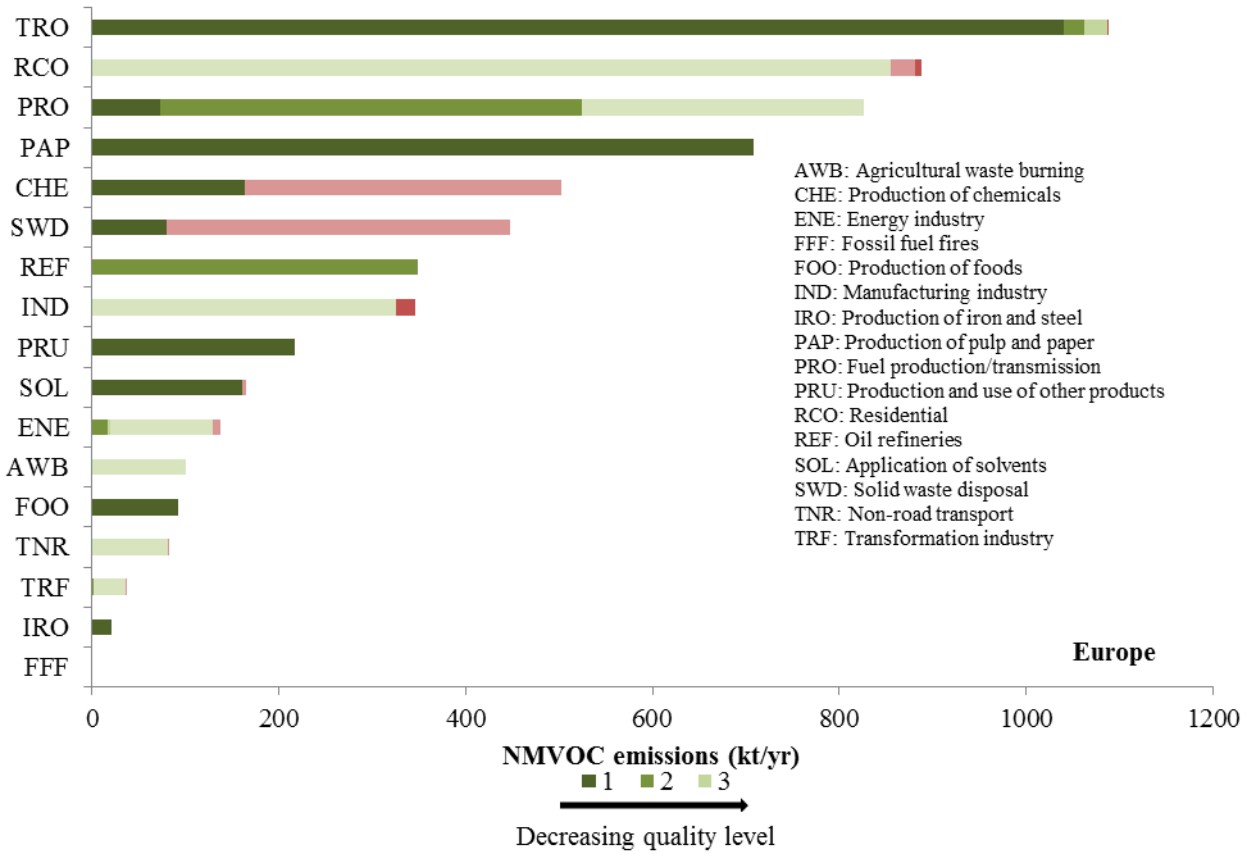

**Figure 11. NMVOC emissions of different sources associated to each quality code in 2010 in Europe.**





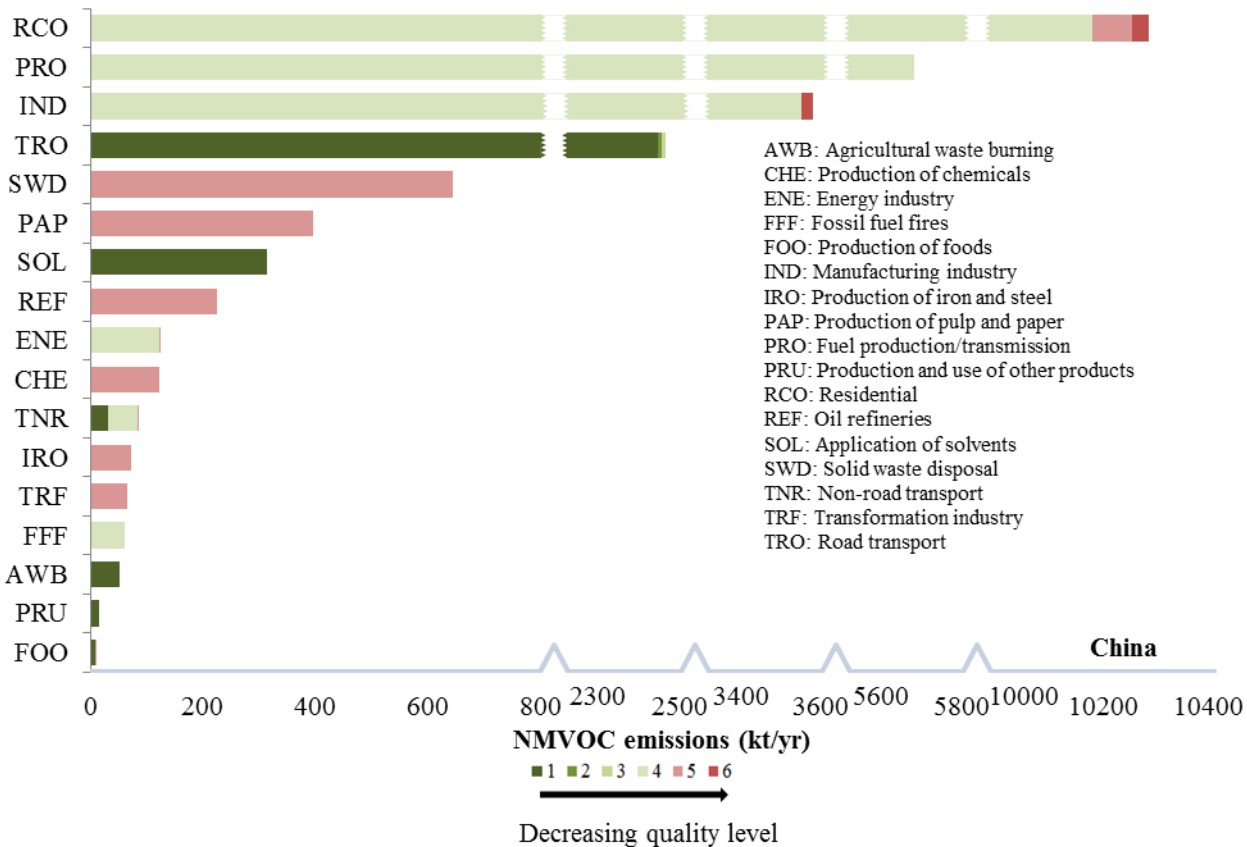

**Figure 12. NMVOC emissions of different sources associated to each quality code in 2010 in China.**





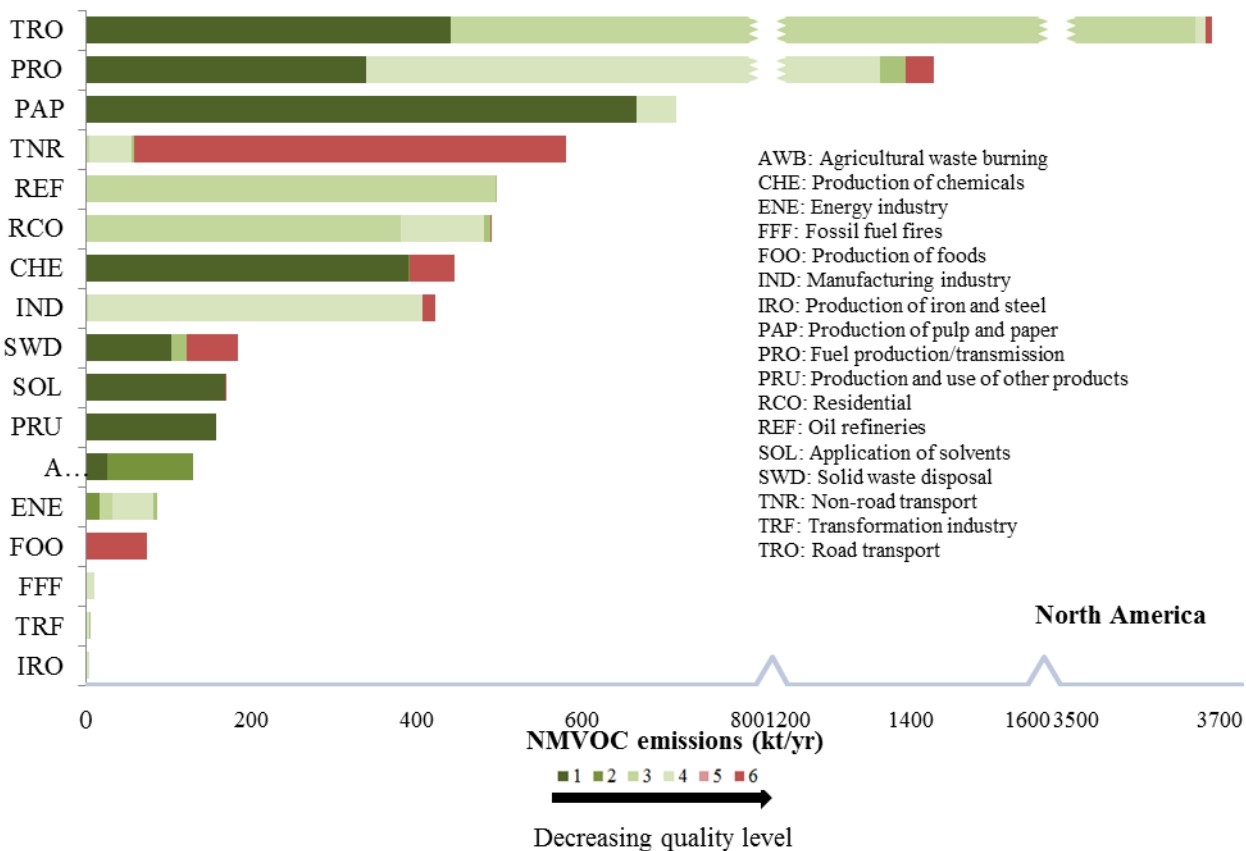

AWB: Agricultural waste burning
CHE: Production of chemicals
ENE: Energy industry
FFF: Fossil fuel fires
FOO: Production of foods
IND: Manufacturing industry
IRO: Production of iron and steel
PAP: Production of pulp and paper
PRO: Fuel production/transmission
PRU: Production and use of other products
RCO: Residential
REF: Oil refineries
SOL: Application of solvents
SWD: Solid waste disposal
TNR: Non-road transport
TRF: Transformation industry
TRO: Road transport

**Figure 13. NMVOC emissions of different sources associated to each quality code in 2010 in North America.**





**Figure 14. Emissions of 25 NMVOC groups associated to each quality code in 2010 in Europe.**




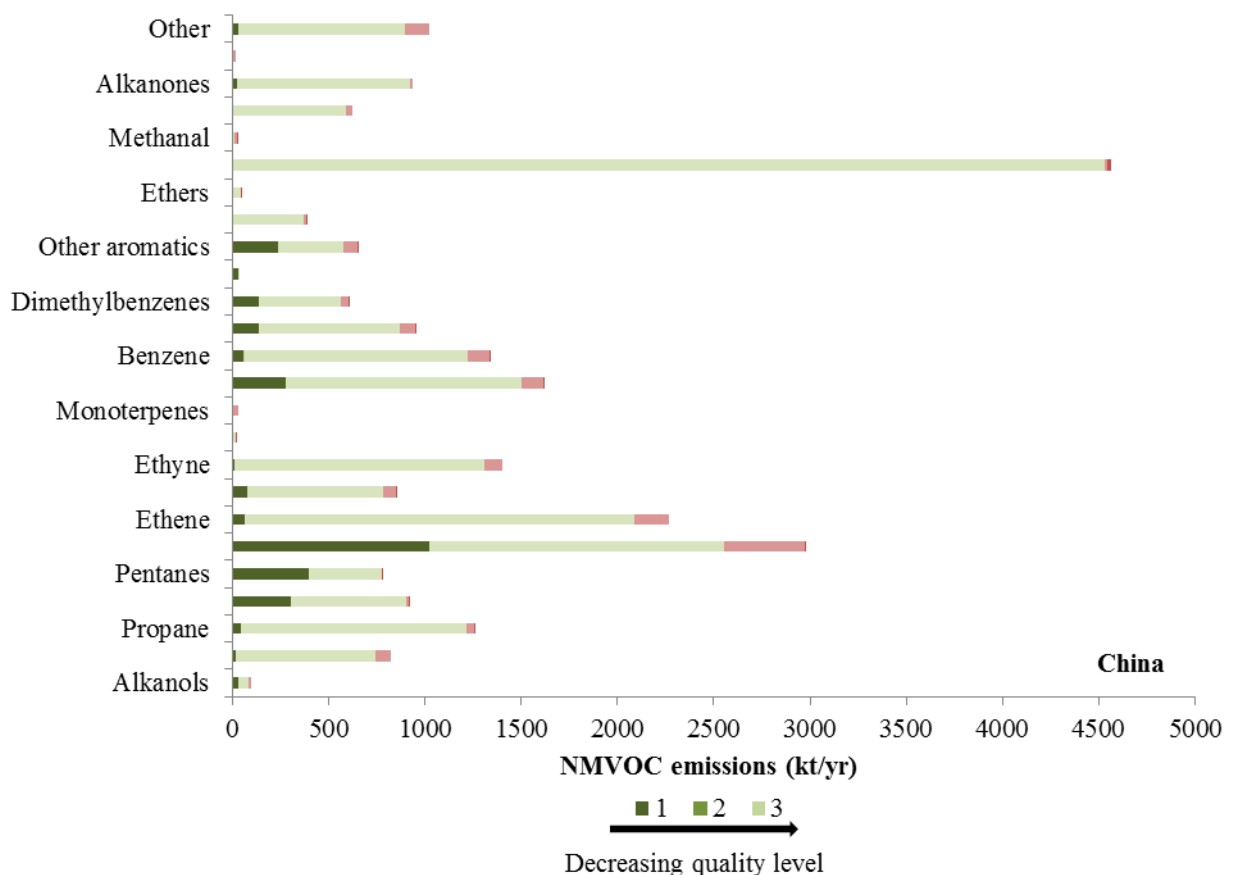

**Figure 15. Emissions of 25 NMVOC groups associated to each quality code in 2010 in China.**



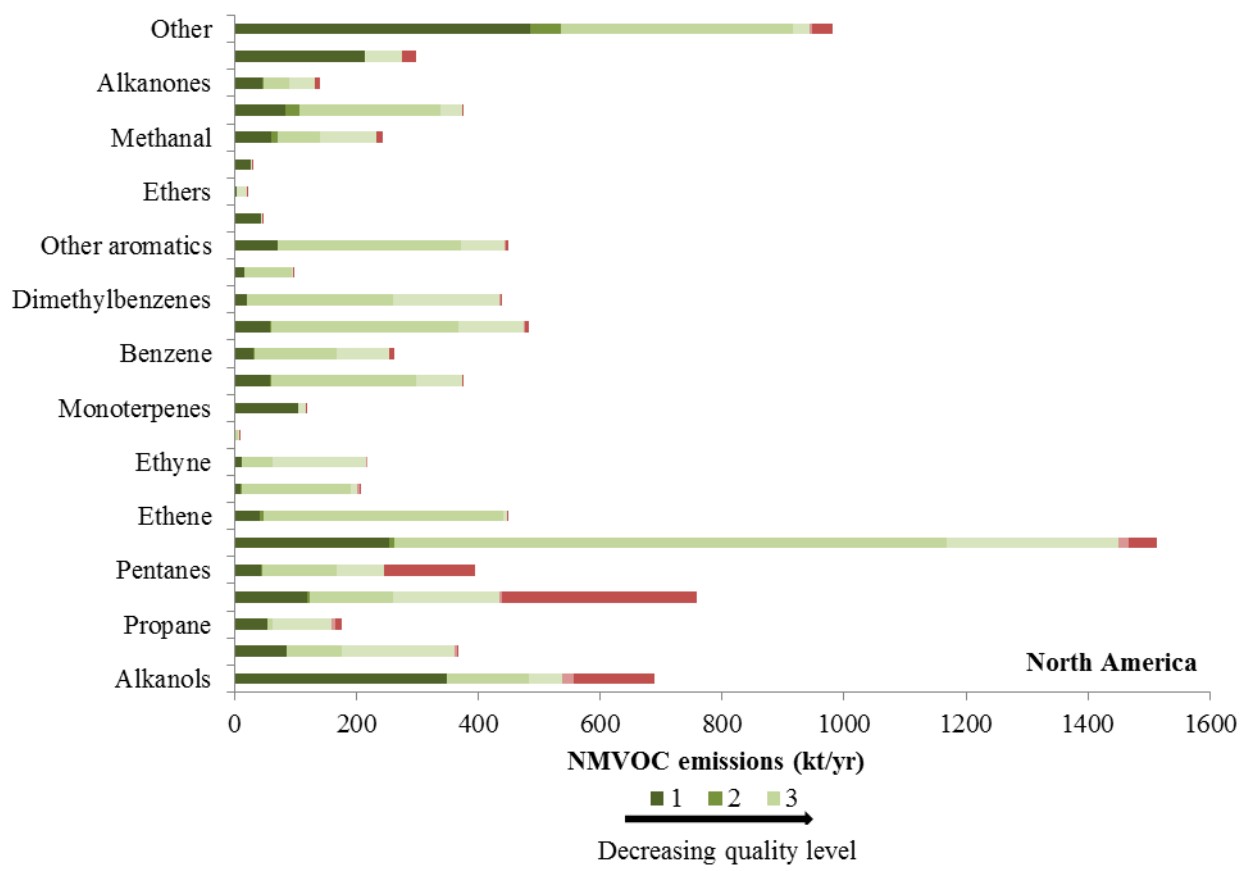

**Figure 16. Emissions of 25 NMVOC groups associated to each quality code in 2010 in North America.**




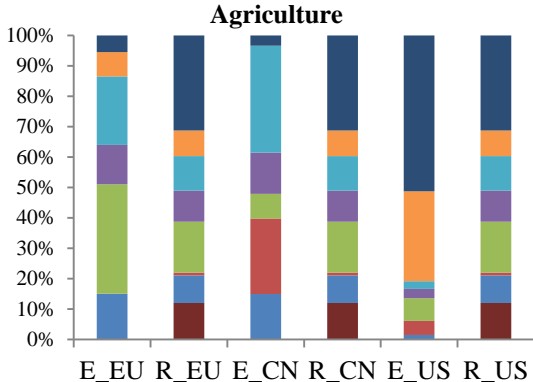

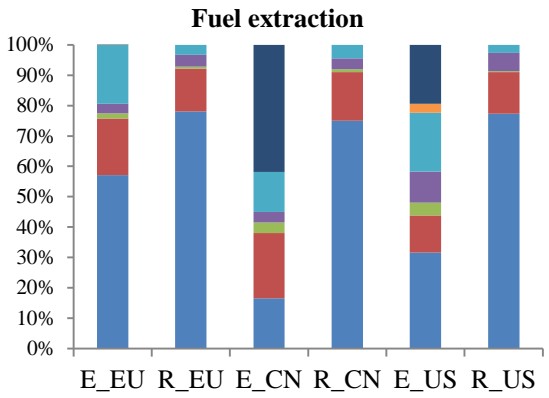

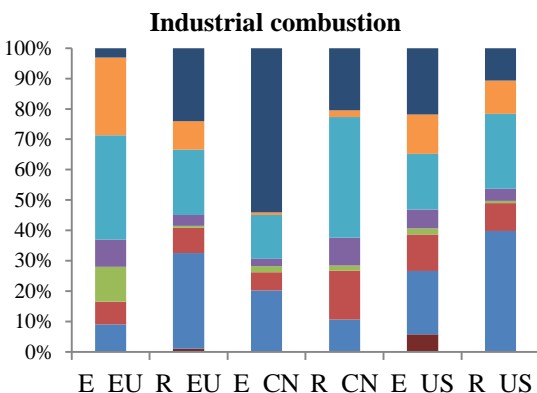

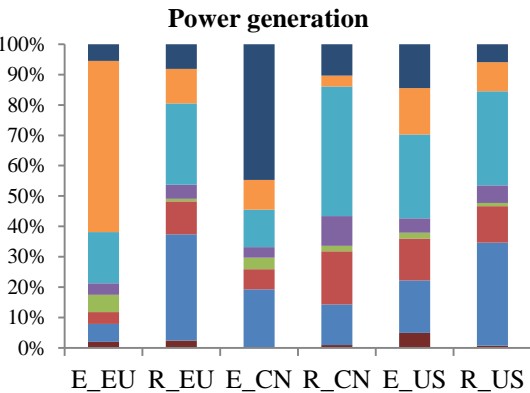

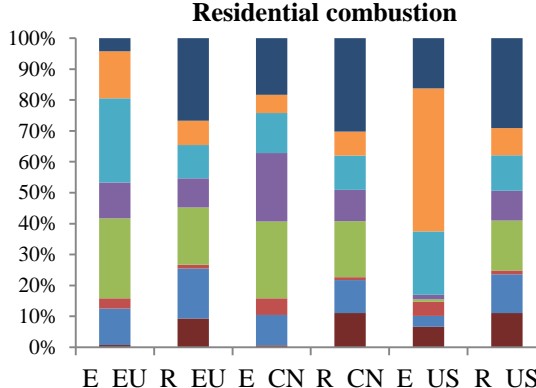

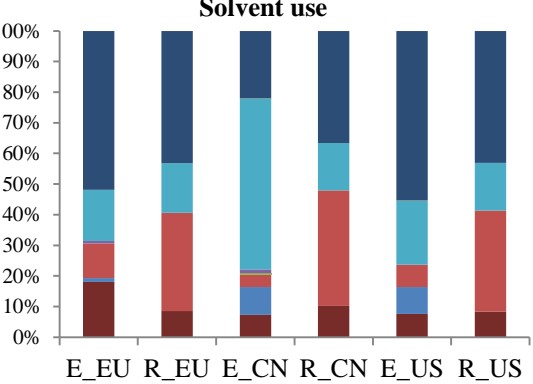





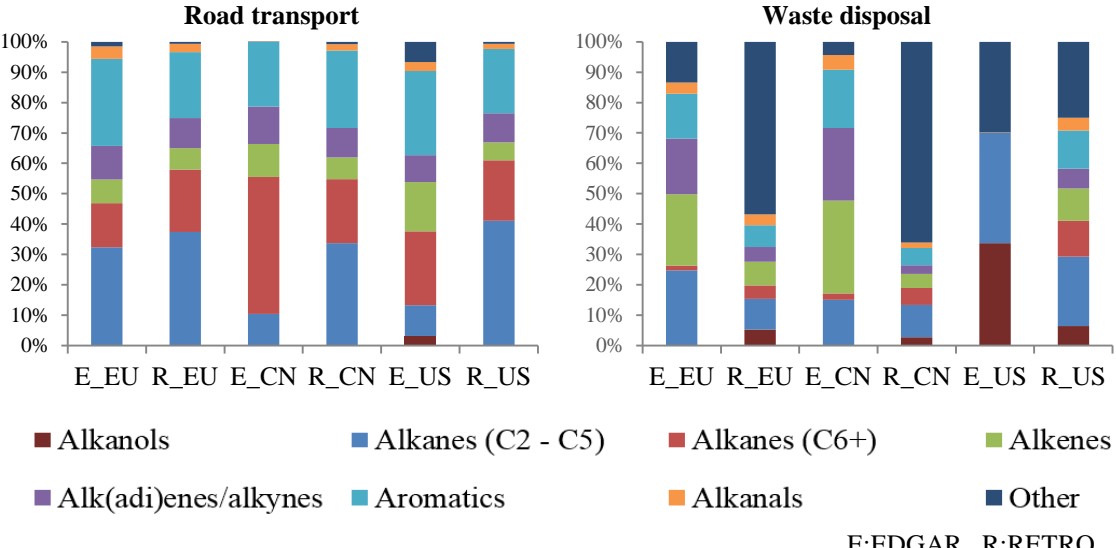

**Figure 17.** Comparison of NMVOC species composition of eight sectors between EDGAR (E) and RETRO (R) data sets for Europe (EU), China (CN) and the United States (US) in 2000.