# Peer review of "Speciation of anthropogenic emissions of non-methane volatile organic compounds: a global gridded data set for 1970-2012"

_Atmospheric Chemistry and Physics, 2017_

## Referee Comment (RC1) · Anonymous Referee #1 · 1 Mar 2017

General comments: The paper presents a global inventory of speciated non-methane volatile organic compounds for the period of 1970 to 2012 based on EDGAR v4.3.2 at a resolution 0.1 x 0.1 degree. This work provides important dataset for global chemical transport model simulation, and give indications on sources and regions where more specific reliable profiles are needed. The manuscript was generally written in a clear way, but more analyses on emission characteristics of other regions except Europe are needed. Detailed emission inventory dataset and profiles used for speciation should be provided, and the large discrepancies of NMVOC emissions with previous studies (especially China) should be illustrated. The manuscript should be carefully checked for figures and text and mistakes should be corrected. I recommend the manuscript to

be revised considering the following comments.

Specific comments: Sect. 2.1: 1. Emissions are grouped into 14 emission sectors, including power generations, industrial combustions etc., which is inconsistent with Table S1 (19 sectors). Please specify the reasons of the sources grouping, since the specification of source classification is key to the profile mapping in the next step. 2. You give detailed description on comparisons between different versions of EDGAR dataset, I don't think it's necessary in the text and there are no relevant discussions in other parts of the manuscript. On the other hand, please give more information on the sources of the raw emission factors, technology assumptions by regions, and abatement measures considered in EDGAR v4.3.2 among world regions. 3. Line 20: please be cautious on the use of "underestimation" when comparing emission inventories. Please check this through manuscript. 4. It's unreasonable that power generation contributes the large differences between EDGAR and HTAP. I think the author means the "relative differences" instead of "absolute differences" because emissions of power generation really small compared to industrial, residential and transportation sectors. The "relative differences" is misleading to readers since the emission contribution of power generation is not important on global scale. Please revise the sentences accordingly. 5. In Figure 1, the emission differences in industry and residential is large and cannot be neglected for DEU, GBR, POL, please explain the reasons and discuss more in the text. 6. How about the emission differences for other countries and regions except Europe, such as Asia and the US? Please add more discussions on comparisons of emissions in Asia and US.

Sect. 2.2: 1. Emission profiles are really important in NMVOC speciation. Please list the mass fractions of the specific profile for each sector for each region. If the table is too large to present, please add an external data link for download. 2. Profiles were measured and developed in various years. Please specify how you apply the profile to sectors in different years and why. Have you considered the trend of the profiles because of the technology evolution? When you assign the quality code in the profile

mapping, have you considered the year when the profiles are measured?

Sect. 2.4: 1. It should be noted that the 25 species groups cannot be directly coupled with CTMs, since individual species are lumped to different chemical mechanisms following different mapping rules. For example, the CB05 mechanism is developed by lumping species according to carbon bond type, while SAPRC-99 is on functional groups. Please specify this clearly in the text. 2. "Where a species contains more than on functional group, priority was typically given to the suffix of the species name since this functional group is generally the most relevant for ozone formation". Please specify clearly what the "suffix of the species name" means. Giving an example here will be better.

Sect. 3.1: 1. Please double check the figure numbers in the manuscript. The figure numbers in the text are inconsistent with the figures. 2. Line 12: "represents" should be "presents"; Line 14: "attribyted" should be "attributed". 3. Please list the Euro standards implemented from 1970 to 2012 as a table in the supplement. 4. Line 24: you mentioned "in addition to aromatics (alkanones, dimethylbenzenes and benzene)...", but alkanones are not aromatics group. Please specify this in the sentence. 5. In the figures, species are grouped to 8 categories: alkanols, alkanes(C2-C5), alkanes(C6+), alkenes, alk(adi)enes/alkynes, aromatics, alkanals, and other. It's not clear how the 25 species mapped to these categories. Please list the mapping process as a table in the text or in the supplement.

Sect. 3.2: 1. The title of Sect. 3.2 is "Case study on the impact of reduction measures on speciated NMVOC emission", but only studies in Germany and the United Kingdom are presented. A paragraph illustrating why you choose Germany and the UK as a case to illustrate the impact of reduction measures is needed. In Asia, I think there are no national control measures implemented before 2010. How about the trend of US? 2. In each case, only residential and road transport results are presented. Please enrich the analyses to include all sectors (power, industry, residential and transport) to give more detailed illustration on the effect of different reduction measures in different

sectors. 3. In the UK case, please explain more on the trend by species groups. Why the emission fraction of alkanes increased rapidly, while aromatics decreased? It's the same reason of the trend in Germany? Please specify clearly in the text. 4. You mentioned "Approximately 90% of NMVOC emissions from road transport attributed to petrol vehicles". Please specify the year of this emission fraction.

Sect. 4.2: 1. The SWD (solid waste disposal) emissions of China are quite high, while SOL (application of solvent) and REF (oil refineries) are incredibly low compared to previous studies in China (INTEX-B, Li et al., 2014, acp). Please specify the reasons of such huge differences.

Sect. 4.3: 1. It surprises me that hexanes, chlorinated hydrocarbons contribute so high to the emissions in Europe, China and North America. Please specify the sources and profiles that relevant with the high hexanes and chlorinated hydrocarbons emissions to these three regions. 2. The emission fractions of the species group differ significantly compared to other studies in China (Li et al., 2014, acp and references therein). Please illustrate the reasons of such differences.

Figures and tables: 1. Figure 4: specify the spatial resolution in the caption. Specify the mapping table from 25 species groups to the 8 categories in the caption. 2. Combining Fig. 6 and Fig. 7 into one figure as (a) and (b) will be better, the same to Fig. 8 and Fig. 9 for UK. 3. Figure 10: the color scale of the quality level is difficult to recognize for reader, especially to distinguish between level 3 and level 4. Use one more distinct color scale. 4. Figure 11: the color legend is not complete. 5. Figure 15 and Figure 16: the Y-axis label (the species name) is not complete.

References: Li, M., Zhang, Q., Streets, D. G., He, K. B., Cheng, Y. F., Emmons, L. K., Huo, H., Kang, S. C., Lu, Z., Shao, M., Su, H., Yu, X., and Zhang, Y.: Mapping Asian anthropogenic emissions of non-methane volatile organic compounds to multiple chemical mechanisms, Atmos. Chem. Phys., 14, 5617-5638, 10.5194/acp-14-5617-2014, 2014.

---

## Referee Comment (RC2) · W. Wei (Referee) · 16 Mar 2017

It is a nice effort to improve the global NMVOC emission database in time, in sector, and in speciation resolution. The data extended by the authors will greatly help the application of VOC emission inventory in the chemical transport simulation of various air quality models. However, the method of the revision of EDGAR NMVOC emission is obscure. The emission factors from EMEP/EEA guidebook were mainly from European references, but their application in developing countries has certain uncertainty. It needs to be further analyzed and evaluated. Moreover, the average abatement efficiencies of the abatement measures for various sectors in various countries should be more introduced in the manuscript. These issues had better be properly handled by

the authors before publication of this paper.

---

## Short Comment (SC1) · 23 Mar 2017

This is potentially a very useful data-set, but I would just like to express my agreement with Ref #1's comment that "Detailed emission inventory dataset and profiles used for speciation should be provided", and to expand on this a little.

I have not read this manuscript in detail (I just came across it), but one comment worried me. At the start of the results section it states: "The compiled NMVOC speciation profiles and their allocation to EDGAR processes and IPCC sectors are available as supplementary data to this article.". This sounded great, but the supplement just contains a lot of graphics; nothing that can be used in practice. I also didn't find this data on the EDGAR web-site. Maybe it will appear one day, but I didn't see any guarantee

of that in the manuscript.

It would be good to see the data made available in computer-readable form (e.g. .csv, netcdf), so that the profiles can be used by those of us that frequently make use of non-EDGAR data. For my own work with the EMEP MSC-W model this would include various HTAP data, ECLIPSE, GAINS, and of course EMEP inventories, which might use SNAP sectors or NFR sectors. Of course, I do not ask that the authors provide data for all inventories, but an easy way of genaralising the results of this study would be to provide Tables of VOC speciation per country and EDGAR sector. Then modellers could easily map these to other gridded data sets.

If such data could be made available it would be a very good addition to the global data-sets which are needed for today's modelling challenges.

Best Regards,

Dave Simpson

EMEP MSC-W, Norwegian Meteorological Institute, Oslo, Norway & Chalmers Univ. Technology, Gothenburg, Sweden

---

## Short Comment (SC2) · 27 Mar 2017

The authors acknowledge the comment of David Simpson regarding the supplementary data provided by this work.

We would like to highlight that as stated in the title of our paper "Speciation of anthropogenic emissions of non-methane volatile organic compounds: a global gridded data set for 1970–2012", the aim of our work is to provide global emission gridmaps over the past 4 decades for NMVOC species and not directly publishing each region- and subsector-specific speciation profile applied to each EDGAR activity code. None of the subsets would provide a comprehensive profile with world coverage directly applicable to a full sector. We want to reassure the Reviewer that the data he is asking for is fully available on the EDGAR website: http://edgar.jrc.ec.europa.eu/overview.php?v=432_VOC_spec&SECURE=123, with for each NMVOC species emission timeseries (1970-2012) by sector and country in an overview table (.xls). Any user can select the IPCC sectors he is interested to look at and calculate the speciation profile by IPCC code and country using the information provided for the 25 NMVOC species and any user can adopt the speciated emissions (gridded and not gridded) to rescale his own NMVOC emission inventory at different level of detail. We use the standard IPCC codes, as these are well defined and any user can convert his/her own sectors to this standard using their own cross-walk matrix. We hope to have clarified the question of David Simpson, as well as having highlighted the possibility for any modeler in applying our speciated database through basic rescaling procedures and VOC species ratios calculations.

---

## Author Comment (AC1) · 21 Apr 2017

Comments from W. Wei (Referee) : It is a nice effort to improve the global NMVOC emission database in time, in sector, and in speciation resolution. The data extended by the authors will greatly help the application of VOC emission inventory in the chemical transport simulation of various air quality models. However, the method of the revision of EDGAR NMVOC emission is obscure. The emission factors from EMEP/EEA guidebook were mainly from European references, but their application in developing countries has certain uncertainty. It needs to be further analyzed and evaluated. Moreover, the average abatement efficiencies of the abatement measures for various sectors in various countries should be more introduced in the manuscript. These issues

had better be properly handled by the authors before publication of this paper.

Response: The authors are grateful to the Referee Wei for the comments received. We tried to improve the paper as requested with more information on the EDGAR methodology in Section S4 of the supplementary material, as reported below:

Total NMVOC emissions from a given sector i in a country C accumulated during a year t are estimated with the following formula (Fig. 1) in the EDGAR database:

EDGAR emission estimates are based on country-specific activity data (AD) for each anthropogenic emission sector i, on which a mix of j technologies (TECH) and a mix of k end-of-pipe measures (EOP) are installed; uncontrolled emission factors (EF) for each sector i and technology j with relative reduction (RED) by abatement measure k are also used in the calculation. The technology mix, (uncontrolled) emission factors and end-of-pipe measures are defined at country-specific, regional, country group (e.g. Annex I/ Non-Annex I), or global level. In particular, NMVOC emission factors are consistent with the EMEP/EEA 2013 Guidebook (EEA, 2013) for Europe and scientific literature has been taken into account to introduce country- and region- specific information, while abatement measures are implemented mainly for the road transport sector (consistent with the Euro standards), for the production of chemicals (CHa-formaldehyde (methanal), total polyethylene, CHa-propylene glycol, total polystyrene), for power generation (auto produced electricity and public electricity production from natural gas) and for landfills. Further details on the EDGAR methodology can be found in Section S4 of the Supplementary material of Crippa et al. (2016).

References

Crippa, M., Janssens-Maenhout, G., Dentener, F., Guizzardi, D., Sindelarova, K., Muntean, M., Van Dingenen, R. and Granier, C.: Forty years of improvements in European air quality: regional policy-industry interactions with global impacts, Atmos. Chem. Phys., 16(6), 3825–3841, doi:10.5194/acp-16-3825-2016, 2016.

EEA: EMEP-EEA emission inventory guidebook – 2013, European Environment Agency. Internet: www.eea.europa.eu/publications, 2013.

[Figure]

$$EM_i(C,t) = \sum_{j,k} \left\lfloor AD_i(C,t) * TECH_{i,j}(C,t) * EOP_{i,j,k}(C,t) * EF_{i,j}(C,t) * \left(1 - RED_{i,j,k}(C,t)\right) \right\rfloor$$

**Fig. 1.** Equation for NMVOC emissions accounting

---

## Author Comment (AC2) · 21 Apr 2017

Answers to Reviewer 1 (Anonymous Referee #1)

General comments: The paper presents a global inventory of speciated non-methane volatile organic compounds for the period of 1970 to 2012 based on EDGAR v4.3.2 at a resolution 0.1 x 0.1 degree. This work provides important dataset for global chemical transport model simulation, and gives indications on sources and regions where more specific reliable profiles are needed. The manuscript was generally written in a clear way, but more analyses on emission characteristics of other regions except Europe are needed. Detailed emission inventory dataset and profiles used for speciation should be provided, and the large discrepancies of NMVOC emissions with previous studies

(especially China) should be illustrated. The manuscript should be carefully checked for figures and text and mistakes should be corrected. I recommend the manuscript to be revised considering the following comments.

Response: The authors thank the referee for the supportive summary and valuable comments towards the improvement of our manuscript. We have addressed each of the referee's comments and revised the manuscript accordingly as elaborated below. The modified parts of the manuscript and supplementary material are highlighted in the revised versions.

Specific comments:

Sect. 2.1:

1. Emissions are grouped into 14 emission sectors, including power generations, industrial combustions etc., which is inconsistent with TableS1 (19 sectors). Please specify the reasons of the sources grouping, since the specification of source classification is key to the profile mapping in the next step.

Response: Table S1 has been removed since it reported the EDGAR activity codes but with a different aggregation compared to what published on the EDGAR NMVOC speciation website. In addition section 2.1 has been rephrased accordingly with the Reviewer's suggestions.

2. You give detailed description on comparisons between different versions of EDGAR dataset, I don't think it's necessary in the text and there are no relevant discussions in other parts of the manuscript. On the other hand, please give more information on the sources of the raw emission factors, technology assumptions by regions, and abatement measures considered in EDGAR v4.3.2 among world regions.

Response: As suggested by the Reviewer, the comparison between different versions of the EDGAR database has been moved to the supplementary material and the following sentences are reported in the main text:

[Figure]

"Figure S1 of the supplementary material shows the comparison of global NMVOC emissions by sector for different EDGAR versions v4.2 (refer to http://edgar.jrc.ec.europa.eu/overview.php?v=42), v4.3.1 (refer to http://edgar.jrc.ec.europa.eu/overview.php?v=431) and v4.3.2 (http://edgar.jrc.ec.europa.eu/overview.php?v=432_VOC_spec&SECURE=123) for the most recent year (2008) available for all datasets. In addition, Figures S2 and S3 show the comparison of NMVOC emissions of EDGARv4.3.2 and the best estimates provided by the HTAP_v2.2 inventory for the year 2010 by HTAP sector and country (refer to Janssens-Maenhout et al. (2015) and http://edgar.jrc.ec.europa.eu/htap_v2/index.php). Focusing on European countries (see Fig. S4), detailed comparison by sector and country (defined with ISO codes) is also performed with officially reported EEA NMVOC emission inventories for the year 2010 (http://www.eea.europa.eu/data-and-maps/data/national-emissions-reported-to-the-convention-on-long-range-transboundary-air-pollution-lrtap-convention-10). "

We tried to improve the paper as requested with more information on the EDGAR methodology in Section S4 of the supplementary material, as reported below:

"Total NMVOC emissions from a given sector i in a country C accumulated during a year t are estimated with the following formula (Fig.1) in the EDGAR database:

EDGAR emission estimates are based on country-specific activity data (AD) for each anthropogenic emission sector i, on which a mix of j technologies (TECH) and a mix of k end-of-pipe measures (EOP) are installed; uncontrolled emission factors (EF) for each sector i and technology j with relative reduction (RED) by abatement measure k are also used in the calculation. The technology mix, (uncontrolled) emission factors and end-of-pipe measures are defined at country-specific, regional, country group (e.g. Annex I/ Non-Annex I), or global level. In particular, NMVOC emission factors are consistent with the EMEP/EEA 2013 Guidebook (EEA, 2013) for Europe and scientific literature has been taken into account to introduce country- and region- specific

information, while abatement measures are implemented mainly for the road transport sector (consistent with the Euro standards), for the production of chemicals (CHa-formaldehyde (methanal), total polyethylene, CHa-propylene glycol, total polystyrene), for power generation (auto produced electricity and public electricity production from natural gas) and for landfills. Further details on the EDGAR methodology can be found in Section S4 of the Supplementary material of Crippa et al. (2016a)".

3. Line 20: please be cautious on the use of "underestimation" when comparing emission inventories. Please check this through manuscript.

Response: Line 20 and the following paragraphs have been rephrased following the Reviewer's suggestion, as reported in section S1 of the supplementary.

4. it's unreasonable that power generation contributes the large differences between EDGAR and HTAP. I think the author means the "relative differences" instead of "absolute differences" because emissions of power generation really small compared to industrial, residential and transportation sectors. The "relative differences" is misleading to readers since the emission contribution of power generation is not important on global scale. Please revise the sentences accordingly.

Response: This section has been rephrased following the Reviewer's suggestion as reported in section S1 of the supplementary.

5. In Figure 1, the emission differences in industry and residential is large and cannot be neglected for DEU, GBR, POL, please explain the reasons and discuss more in the text.

Response: The description of Figure 1 (now figure S4) has been modified as following:

Focusing on European countries (see Fig. S4), detailed comparison by sector and country (defined with ISO codes) is also performed with officially reported EEA NMVOC emission inventories for the year 2010 (http://www.eea.europa.eu/data-and-maps/data/national-emissions-reported-to-the-convention-on-long-range-

transboundary-air-pollution-lrtap-convention-10). Total NMVOC emissions at European scale are 15% higher for EDGAR compared to EEA and HTAP_v2.2. However, insights on the origin of such differences can be retrieved looking at sectorial emissions. The power generation sector in EU represents less than 2% of total NMVOC emissions although it shows quite some discrepancies among inventories. As shown in Fig. S3 and Fig. S4, industrial, residential and ground transport NMVOC emissions are characterized by better agreement among the three inventories, with the exception of few countries. EDGAR estimates 30-50% lower emissions for ground transport emissions for France, Poland and Czech Republic compared to HTAP and EEA, while it generally overestimates residential emissions (e.g. in particular for Germany, France and UK, possibly due to an underestimation of the combustion of biomass in the household sector as reported by van der Gon et al. (2015)). Differences in the NMVOC emissions of the industrial sector among the inventories might be due to the underestimation by 50% of the EDGAR gas distribution subsector for Europe and by 15% at the global scale.

6. How about the emission differences for other countries and regions except Europe, such as Asia and the US? Please add more discussions on comparisons of emissions in Asia and US.

Response: Comparison of 2010 NMVOC sectorial emissions estimated by EDGARv4.3.2 and HTAP_v2 for Asian countries and North America are reported in Figure S2 and the following description is already reported in the supplementary material. More detailed comparisons are beyond the scope of this paper.

Figures S2 and S3 show the comparison of NMVOC emissions of EDGARv4.3.2 and the best estimates provided by the HTAP_v2.2 inventory for the year 2010 by HTAP sector and country (refer to Janssens-Maenhout et al. (2015) and http://edgar.jrc.ec.europa.eu/htap_v2/index.php). Very good agreement for all sectors is found between EDGARv4.3.2 and HTAP_v2.2 for Asian countries and North America (refer to Fig. S2), as well as for Europe (refer to Fig. S3). Lower NMVOC emissions

are reported by EDGARv4.3.2 for India and Indonesia for the residential and transport sectors compared to the HTAPv2 data (although the reported HTAP_v2.2 emissions appear to be very high compared for example with the Chinese ones).

Sect. 2.2:

1. Emission profiles are really important in NMVOC speciation. Please list the mass fractions of the specific profile for each sector for each region. If the table is too large to present, please add an external data link for download.

Response: Thanks to the reviewer's interest in the data. We would like to highlight that as stated in the title of our paper "Speciation of anthropogenic emissions of non-methane volatile organic compounds: a global gridded data set for 1970–2012", the aim of our work is to provide global emission gridmaps over the past 4 decades for NMVOC species and not directly publishing each region- and subsector-specific speciation profile applied to each EDGAR activity code. None of the subsets would provide a comprehensive profile with world coverage directly applicable to a full sector. We want to reassure the Reviewer that the data he is asking for is fully available on the EDGAR website: http://edgar.jrc.ec.europa.eu/overview.php?v=432_VOC_spec&SECURE=123, with for each NMVOC species emission time series (1970-2012) by sector and country in an overview table (.xls). Any user can select the IPCC sectors he is interested to look at and calculate the speciation profile by IPCC code and country using the information provided for the 25 NMVOC species and any user can adopt the speciated emissions (gridded and not gridded) to rescale his own NMVOC emission inventory at different level of detail. We use the standard IPCC codes, as these are well defined and any user can convert his/her own sectors to this standard using their own cross-walk matrix. We hope to have clarified this request, as well as have highlighted the possibility for any modeller in applying our speciated database through basic rescaling procedures and VOC species ratios calculations.
2. Profiles were measured and developed in various years. Please specify how you apply the profile to sectors in different years and why. Have you considered the trend of the profiles because of the technology evolution? When you assign the quality code in the profile mapping, have you considered the year when the profiles are measured?

Response: Speciation profiles were mapped to all EDGAR process codes, which have a very high sector resolution differentiating source group, sector, fuel type, technology and end-of-pipe measures related to NMVOC emissions. Technological evolution is reflected by assigning technological differentiated profiles to specific process codes, e.g. different profiles for emissions from conventional or closed-loop-catalyst gasoline vehicles. When technological specific profiles are not available, this is reflected in the assigned quality code.

Most of the collected profiles, e.g. from the SPECIATE database, does not provide any information on the year of the profiles, which is identified as a limitation but it should be recognised that the best data available has been utilised. Moreover, differences of profiles measured in various year could be partly attributed to technological evolution.

Sect. 2.4:

1. It should be noted that the 25 species groups cannot be directly coupled with CTMs, since individual species are lumped to different chemical mechanisms following different mapping rules. For example, the CB05 mechanism is developed by lumping species according to carbon bond type, while SAPRC-99 is on functional groups. Please specify this clearly in the text.

Response: Thanks to this note. The text is amended accordingly. Moreover, it is worth noting that speciated data is available at the most detailed level for those that wish to obtain it and then perform their own aggregation.

2. "Where a species contains more than on functional group, priority was typically given to the suffix of the species name since this functional group is generally the most

relevant for ozone formation". Please specify clearly what the "suffix of the species name" means. Giving an example here will be better.

Response: We have added an example as following, and hope it clarifies the confusion.

For example, trichlorobenzenes are assigned to "other aromatics" rather than "chlorinated hydrocarbons" as the suffix of the species name belongs to the aromatics group.

Sect. 3.1:

1. Please double check the figure numbers in the manuscript. The figure numbers in the text are inconsistent with the figures. 2. Line 12: "represents" should be "presents"; Line 14: "attribyted" should be "attributed". 3. Please list the Euro standards implemented from 1970 to 2012 as a table in the supplement. 4. Line 24: you mentioned "in addition to aromatics (alkanones, dimethylbenzenes and benzene)...", but alkanones are not aromatics group. Please specify this in the sentence. 5. In the figures, species are grouped to 8 categories: alkanols, alkanes (C2-C5), alkanes(C6+), alkenes, alk(adi)enes/alkynes, aromatics, alkanals, and other. It's not clear how the 25 species mapped to these categories. Please list the mapping process as a table in the text or in the supplement.

Response: Figure numbers and text corrections have been implemented as suggested by the Reviewer. In addition, a table mapping the 25 NMVOC species to 8 categories has been introduced in the supplementary material as well as a table with the Euro standard implementation as available for the EDGAR database.

Sect. 3.2:

1. The title of Sect. 3.2 is "Case study on the impact of reduction measures on speciated NMVOC emission", but only studies in Germany and the United Kingdom are presented. A paragraph illustrating why you choose Germany and the UK as a case to illustrate the impact of reduction measures is needed. In Asia, I think there are no national control measures implemented before 2010. How about the trend of US? 2. In

each case, only residential and road transport results are presented. Please enrich the analyses to include all sectors (power, industry, residential and transport) to give more detailed illustration on the effect of different reduction measures in different sectors.

Response: We have modified this section following the reviewer's suggestions. The US is much more complicated compared to e.g. the UK, due to differences in state and federal laws, similar for Asia. We agree that an analysis on the impact of reduction measures on speciated NMVOC emissions covering all sectors and main regions would be very interesting. However few data are available. We have added some discussion for industry and other sectors in the UK. More detailed analyses are beyond the scope of this paper.

3. In the UK case, please explain more on the trend by species groups. Why the emission fraction of alkanes increased rapidly, while aromatics decreased? It's the same reason of the trend in Germany? Please specify clearly in the text. 4. You mentioned "Approximately 90% of NMVOC emissions from road transport attributed to petrol vehicles". Please specify the year of this emission fraction.

Response: Modification of the text has been made as shown in the revised paper following the reviewer's comments.

Sect. 4.2:

1. The SWD (solid waste disposal) emissions of China are quite high, while SOL (application of solvent) and REF (oil refineries) are incredibly low compared to previous studies in China (INTEX-B, Li et al., 2014, acp). Please specify the reasons of such huge differences.

Response: We checked and updated the data and figures with a revised version of the EDGARv4.3.2 database in particular for the solvent use and waste sectors. Figure and corresponding text are modified in the revised manuscript. The contributions of residential sources to NMVOC emissions in china are comparable according to EDGAR

(20% in 2010) and INTEX-B (24.1% in 2006). However, the categorization of sectors of EDGAR and INTEX-B (reported in Li et al. 2014) are quite different, which makes a direct comparison difficult. For example solvent use is not reported as an individual source in Li et al. 2014, but classified to sectors like residential non-combustion and industrial non-combustion.

Sect. 4.3:

1. It surprises me that hexanes, chlorinated hydrocarbons contribute so high to the emissions in Europe, China and North America. Please specify the sources and profiles that relevant with the high hexanes and chlorinated hydrocarbons emissions to these three regions. 2. The emission fractions of the species group differ significantly compared to other studies in China (Li et al., 2014, acp and references therein). Please illustrate the reasons of such differences.

Response: In order to address the reviewer's comments we added some discussion in section 4.3 about the contributing sources and profiles of the most abundant specie groups in the three regions respectively. We conducted a preliminary comparison of our results with Li et al., 2014. The two studies show agreement on the abundances of dimethylbenzenes, methylbenzene, benzene and ethene of NMVOC emissions in China. High emission levels of chlorinated hydrocarbons and hexanes and higher alkanes found in this study is not addressed in Li et al., 2014. This could probably be owing to on one hand different categorization of NMVOC species groups, and on the other hand the adoption of different speciation profiles. We have discussed in the revised manuscript the sources and profiles related to chlorinated hydrocarbons and hexanes and higher alkanes emissions in China.

Figures and tables: 1. Figure 4: specify the spatial resolution in the caption. Specify the mapping table from 25 species groups to the 8 categories in the caption. 2. Combining Fig. 6 and Fig. 7 into one figure as (a) and (b) will be better, the same to Fig. 8 and Fig. 9 for UK. 3. Figure 10: the color scale of the quality level is difficult to recognize

for reader, especially to distinguish between level 3 and level 4. Use one more distinct color scale. 4. Figure 11: the color legend is not complete. 5. Figure 15 and Figure16: the Y-axis label (the species name) is not complete.

Response: Many thanks for the reviewer's careful reading. The caption of figure 4 has been modified following the reviewer's suggestion as following:

Figure 4. NMVOC emission gridmap at 0.1x0.1 degree resolution from the residential sector in 2010. The relative contribution of 8 aggregated NMVOC species is reported in the pie charts for major world regions (number in brackets refer to total NMVOC emissions (in ktons) for the residential sector for each region).

Figures of case study for Germany and the UK have been combined as suggested by the reviewer. The other noted figures are also modified accordingly.

References

Janssens-Maenhout, G., Crippa, M., Guizzardi, D., Dentener, F., Muntean, M., Pouliot, G., Keating, T., Zhang, Q., Kurokawa, J., and Wankmüller, R.: HTAP_v2. 2: a mosaic of regional and global emission grid maps for 2008 and 2010 to study hemispheric transport of air pollution, Atmospheric Chemistry and Physics, 15, 11411-11432, 2015.

Li, M., Zhang, Q., Streets, D. G., He, K. B., Cheng, Y. F., Emmons, L. K., Huo, H., Kang, S. C., Lu, Z., Shao, M., Su, H., Yu, X. and Zhang, Y.: Mapping Asian anthropogenic emissions of non-methane volatile organic compounds to multiple chemical mechanisms, Atmos. Chem. Phys., 14(11), 5617–5638, doi:10.5194/acp-14-5617-2014, 2014.

Please also note the supplement to this comment:
http://www.atmos-chem-phys-discuss.net/acp-2017-65/acp-2017-65-AC2-supplement.zip

$$EM_i(C,t) = \sum_{j,k} \left[ AD_i(C,t) * TECH_{i,j}(C,t) * EOP_{i,j,k}(C,t) * EF_{i,j}(C,t) * \left(1 - RED_{i,j,k}(C,t)\right) \right]$$

**Fig. 1.** Equation for NMVOC emissions accounting